# *Failure-Proof* NON-CONTRASTIVE SELF-SUPERVISED LEARNING

## ABSTRACT

We identify sufficient conditions to avoid known failure modes, including representation, dimensional, cluster and intracluster collapses, occurring in non-contrastive self-supervised learning. Based on these findings, we propose a principled design for the projector and loss function. We theoretically demonstrate that this design introduces an inductive bias that promotes learning representations that are both decorrelated and clustered without explicit enforcing these properties and leading to improved generalization. To the best of our knowledge, this is the first solution that achieves robust training with respect to these failure modes while guaranteeing enhanced generalization performance in downstream tasks. We validate our theoretical findings on image datasets including SVHN, CIFAR10, CIFAR100 and ImageNet-100, and show that our solution, dubbed *FALCON*, outperforms existing feature decorrelation and cluster-based self-supervised learning methods in terms of generalization to clustering and linear classification tasks.

## 1  INTRODUCTION

Self-supervised learning (SSL) has unlocked the potential of learning general-purpose representations from large amounts of unlabeled data. Despite its successes, important challenges remain, hindering the applicability of SSL to a broader spectrum of real-world tasks and its widespread adoption and democratization. One such challenge is the presence of failure modes occurring during the training of SSL models. Several heuristic strategies have been proposed and analyzed in the literature, such as momentum encoder, stop gradient and asymmetric projector heads (Chen et al., 2022; He et al., 2020; Grill et al., 2020; Tao et al., 2022; Chen & He, 2021; Tian et al., 2021; Halvagal et al., 2023; Wang et al., 2022). However, these heuristics do not always come with universal guarantees, making it unclear whether failure modes can be avoided in all situations.

In this work, we focus on the family of non-contrastive SSL approaches, aiming to distill the essential principles to guarantee the avoidance of known failure modes. More concretely, we identify sufficient conditions to avoid representation, dimensional, cluster, and intracluster collapses, and correspondingly devise a projector and loss function enforcing them by design. In particular, we demonstrate that minimizing invariance to data augmentations while matching priors suffices to avoid representation and cluster collapses, whereas normalized embeddings, orthogonal frozen weights, and large prediction outputs in the projector are key to avoiding dimensional and intracluster collapses. Moreover, we prove that these principles are sufficient to guarantee (i) the decorrelation of embeddings, without any explicit computation of their covariance matrix, and (ii) the clustering of embeddings, without the use of specialized clustering layers. We experimentally validate the theory on four image datasets, including SVHN, CIFAR-10, CIFAR-100 and ImageNet-100, showcasing superior performance in terms of training robustness to failure modes and generalization to downstream clustering and classification tasks compared to popular feature decorrelation and cluster-based SSL.

To summarize, our key contributions are:

- At a conceptual level, we provide sufficient conditions to avoid known failure modes and accordingly devise *FALCON*, the first *FA*i*L*ure-proof non-*CON*trastive SSL approach.
- These conditions are shown to be sufficient to jointly decorrelate and cluster embeddings, thus providing evidence on the feasibility of unifying non-contrastive and cluster-based SSL families of approaches.

- We establish the first connection between SSL and hyperdimensional computing, thus supporting training for large projector outputs (reaching similar order of magnitudes to the size of the dataset).
- At a practical and computational level, we simplify the design and training of non-contrastive SSL and provide a proof-of-concept demonstration of the properties of *FALCON*.

The structure of the paper is organized as follows: In §2, we relate our work to theory of SSL, clarify the relation between different failure modes and review recent efforts aiming to address them. In §3, we provide design principles for the loss and projector head of *FALCON*. Subsequently, we provide the theory supporting its design. In §4, we compare our solution to existing feature decorrelation and cluster-based SSL strategies and analyze the different properties highlighted by the theory. Finally, we conclude with §5 by summarizing the main findings and discussing future work.

## 2 RELATED WORK

We frame this work within the context of theoretical studies of SSL and solutions aimed at mitigating collapses.

**Theory and relations among different families of SSL**. Several works have theoretically investigated contrastive (Saunshi et al., 2019; Wang & Isola, 2020; Zimmermann et al., 2021; Tosh et al., 2021; HaoChen et al., 2021; Saunshi et al., 2022; Wang et al., 2024) and non-contrastive SSL methods (Tian et al., 2021; Kang-Jun et al., 2022; Weng et al., 2022; Wen & Li, 2022; Shwartz-Ziv et al., 2023), to improve our understanding and provide more principled or simplified solutions. There have been works identifying key properties for SSL objectives and inductive biases (Wang & Isola, 2020; Dubois et al., 2022), generalizing SSL to incorporate data augmentation graphs (HaoChen et al., 2021; Wang et al., 2024), deriving generalization error bounds (Saunshi et al., 2022; Bao et al., 2022; Shwartz-Ziv et al., 2023), understanding SSL objectives from an optimization perspective (Tian et al., 2021; Tian, 2022; 2023) as well as understanding the role of asymmetries and projector heads (Kang-Jun et al., 2022; Wen & Li, 2022). A recent line of studies has focused on identifying connections between contrastive and non-contrastive methods (Garrido et al., 2023b; Balestriero & LeCun, 2022; Huang et al., 2023) aiming towards unifying different families of SSL. Our work complements these efforts by providing a principled solution and design to bring together cluster-based SSL and feature decorrelation methods from the non-contrastive family.

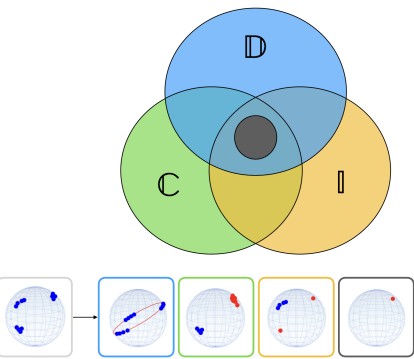

Figure 1: **Top:** Venn diagram relationships between types of collapses, viz. dimensional collapse ($\mathbb{D}$), cluster collapse ($\mathbb{C}$), intracluster collapse ($\mathbb{I}$) and representation (full) collapse of embeddings. **Bottom:** on the left, *failure-proof* data embeddings and on the right, examples of collapses for each single type (collapsed clusters are highlighted in red).

**Failure modes in SSL**. SSL can be affected by four undesired failure modes, namely representation, dimensional, cluster and intracluster collapses, as exemplified in Figure 1. *Representation collapse* refers to the case where neural representations collapse to an identical constant vector, irrespectively of their input. Different strategies have been proposed to avoid the issue, such as leveraging contrastive objectives to maximize mutual information between data and representations (den Oord et al., 2018; O. Henaff, 2020; Chen et al., 2020; Lee, 2022; Linsker, 1988; Becker & Hinton, 1992; McAllester & Stratos, 2020; Barber & Agakov, 2004; Belghazi et al., 2018; Poole et al., 2019; Tschannen et al., 2019; Song & Ermon, 2020; directly estimating and maximizing the mutual information of cluster predictions (X. Ji & J. F. Henriques, 2019), introducing heuristics such as momentum encoder, stop gradient and asymmetric projector heads (Chen et al., 2022; He et al., 2020; Grill et al., 2020; Tao et al., 2022; Chen & He, 2021; Tian et al., 2021; Halvagal et al., 2023; Wang et al., 2022), regularizing the objective by introducing a generative term to reconstruct or estimate the data density (Hendrycks et al., 2019; Winkens et al., 2020; Mohseni et al., 2020; Kim & Ye, 2022; Gatopoulos & Tomczak, 2020; Zhue et al., 2020; Wu et al., 2023; Nakamura et al., 2023; Sansone, 2023; Sansone & Manhaeve, 2024) and leveraging predictive models to identify masked data such as image patches or text tokens (et al., 2022; Zhou et al., 2022). *Dimensional collapse* occurs when embeddings span a subspace of the whole vector space. Several methods (Zbontar et al., 2021; Zhang et al., 2021; Ermolov et al.,

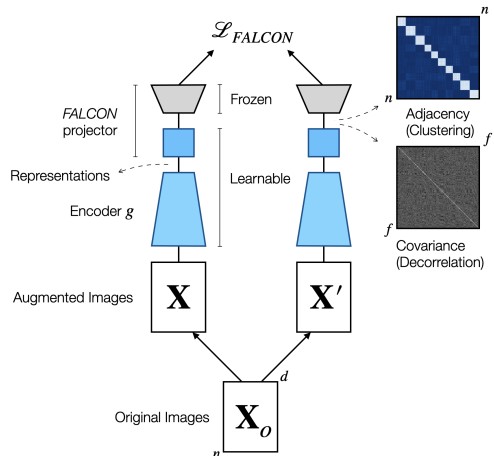

Figure 2: In *FALCON*, minimizing the proposed objective together with the corresponding projector ensures that the embedding representations are clustered and at the same that their features are decorrelated. This guarantees that the representations are *failure-free*, meaning that dimensional, cluster, intra-cluster and representation collapses are prevented.

**Algorithm 1** Pseudocode for *FALCON*

```
# g: encoder network
# n: batch size
# f: embedding dimensionality
# c: dictionary size
# eps: 1e-6
# bn: batch normalization
# norm: L2 normalization activation
# softmax: softmax activation with temperature
# beta: weight for the invariance loss
# ce: crossentropy loss

# compute non-learnable dictionary codes
W = 2 * randint(2, size=(f, c)) - 1 # f-by-c
for X_o in loader: # load a batch with n samples
    # two randomly augmented versions of X_o
    X, X` = augment(x_o)
    # compute representations
    Z = g(X)   # n-by-f
    Z`= g(X`) # n-by-f

    # extract embeddings (blue square block)
    H = norm(bn(linear(Z)))*sqrt(f/n)   # n-by-f
    H`= norm(bn(linear(Z`)))*sqrt(f/n) # n-by-f
    # compute probabilities (gray block)
    tau = f/(sqrt(n) * log((1-eps*(c-1)) / eps))
    P = softmax(H @ W, tau)   # n-by-c
    P`= softmax(H` @ W, tau) # n-by-c
    # compute losses
    loss_prior = ce(1/c, P.mean(0))
    loss_inv = ce(P, P`).mean()
    loss = beta*loss_inv + loss_prior

    # optimization step
    loss.backward()
    optimizer.step()
```

2021; Li et al., 2022b; Liu et al., 2022; Bardes et al., 2022a;b; Ozsoy et al., 2022) propose to mitigate the issue by whitening the feature embeddings (Hua et al., 2021). Dimensional collapse has been recently linked to reduced performance of downstream tasks (He & Ozay, 2022; Li et al., 2022a; Garrido et al., 2023a), and different evaluation metrics have been consequently derived, such as the computation of the entropy of the singular value distribution for the covariance matrix (Jing et al., 2022), the rank estimator (Garrido et al., 2023a), the computation of the AUC (Li et al., 2022a) or a power law approximation (Ghosh et al., 2022) of the singular value distribution. *Cluster collapse* is a phenomenon observed in cluster-based SSL, where data points are assigned to a subset of available prototypes (Caron et al., 2018; Asano et al., 2020; Caron et al., 2020; Li et al., 2021; Caron et al., 2021; Govindarajan et al., 2023). The issue is typically mitigated by introducing regularizers in the objective, such as the Koleo regularizer (Sablayrolles et al., 2019; et al., 2024; Govindarajan et al., 2024) or explicitly enforcing uniform cluster assignments (Amrani et al., 2022; et al., 2023). Last but not least in terms of importance, *intracluster collapse*, similarly to the notion of neural collapse observed in supervised learning (Papyan et al., 2020; Fang et al., 2021; Yang et al., 2022; Chen et al., 2022; Kothapalli, 2023; Dhuliawala et al., 2023) occurs whenever the variability of the embeddings within some clusters is infinitesimally small. Intracluster collapse can be mitigated by enforcing representation equivariance (rather than invariance) to data augmentations (Dangovski et al., 2022; Komodakis & Gidaris, 2018; Scherr et al., 2022; Park et al., 2022) or by splitting the embeddings into content and style parts, while using only content for the self-supervision task (Louizos et al., 2016; Kügelgen et al., 2021; Garrido et al., 2023c). In contrast, this work provides a principled yet simple solution based on an objective function and projector head that avoid all forms of collapses, consequently leading to significant performance gains.

## 3 *FALCON* METHOD AND PROPERTIES

**Notation.** We denote matrices using capital bold letters, e.g. $\mathbf{P}$, their elements using lowercase letters with subscript indices, e.g. $p_{ij}$, their row and column vectors using lowercase bold letters, e.g. $\mathbf{p}_i$ and $\mathbf{p}_j$. Additionally, we denote sets using capital letters, e.g. $\mathcal{S}$ and use squared brackets when dealing with sets of integers, e.g. $[n] \equiv \{1, \ldots, n\}$. Finally, we use lowercase letters for functions, scalars, integers and constants, e.g. $n$. Whenever evaluating functions on matrices, we always assume that the function is applied row-wise.

**Overview of *FALCON*.** Given an unlabeled batch of data $\mathcal{D} = \{(\mathbf{X}, \mathbf{X}')\}$ containing $n$ pairs of augmented images, so that $\mathbf{X}, \mathbf{X}' \in \mathbb{R}^{n \times d}$, we propose to train a backbone encoder $g : \mathbb{R}^d \to \mathbb{R}^f$ using the *FALCON* projector and loss functions.[1] The projector takes the representations $(\mathbf{Z}, \mathbf{Z}') = (g(\mathbf{X}), g(\mathbf{X}'))$, with $\mathbf{Z}, \mathbf{Z}' \in \mathbb{R}^{n \times f}$, and performs two operations. Firstly, it computes embeddings $\mathbf{H}, \mathbf{H}' \in \mathbb{R}^{n \times f}$ for the corresponding representations and then it computes probabilities $\mathbf{P}, \mathbf{P}' \in \mathbb{R}^{n \times c}$ for assigning embeddings to codes available from a frozen dictionary $\mathbf{W} \in \{-1, 1\}^{f \times c}$. More precisely, the projector is defined by the following two layers:

$$\mathbf{H} = \sqrt{f/n} \cdot \text{L2-norm}(\text{Bn}(\text{Linear}(\mathbf{Z})))$$
$$\mathbf{P} = \text{Softmax}(\mathbf{HW}/\tau) \tag{1}$$

where the embeddings are obtained from representations through the composition of linear, batch norm and L2 normalization layers, and $\tau$ is the temperature parameter of the softmax layer. Each element $w_{ij}$ of $\mathbf{W}$ is drawn independently and identically distributed according to a Rademacher distribution, i.e. $w_{ij} \sim$ Rademacher for all $i \in [f]$ and $j \in [c]$, whereas

$$\tau = \frac{f}{\sqrt{n} \log\left(\frac{1 - \epsilon(c-1)}{\epsilon}\right)}$$

with $\epsilon$ an arbitrarily small positive scalar.[2] Notably, we will provide theoretical justification for the design choice of the *FALCON* projector, demonstrating good properties for the embeddings, as being both well clustered and having their features decorrelated. Briefly, the linear and batch norm layers are used to ensure well-behaved statistics throughout training, such as zero mean and unit element variance (cf. Appendix A), while the normalization layer promote embeddings that are aligned with the frozen codes in $\mathbf{W}$ (cf. Th. 1). Moreover, we suggest to choose $c \gg f$ to avoid dimensional and intracluster collapses, as we will show in §3.2 and §3.3. The *FALCON* loss consists of two terms, including one to promote invariance to data augmentations and one for prior matching, namely:

$$\mathcal{L}_{FALCON}(\mathcal{D}) = -\frac{\beta}{n} \sum_{i=1}^{n} \sum_{j=1}^{c} p_{ij} \log p'_{ij} - \sum_{j=1}^{c} q_j \log \frac{1}{n} \sum_{i=1}^{n} p_{ij} \tag{2}$$

with $\mathbf{q} = [q_1, \ldots, q_c] \in \mathbb{Q}^c$ corresponding to a prior probability vector, chosen uniformly for all $c$ codes in all our experiments, viz. $q_j = 1/c$ for all $j \in [c]$, $\beta > 0$ is a weight hyperparameter to balance the relative importance of the two loss terms and $p_{ij}, p'_{ij}$ are elements of $\mathbf{P}, \mathbf{P}'$, respectively. We will prove in §3.1 that, when the two loss terms are minimized, the proposed loss function is guaranteed to avoid representation and cluster collapses and therefore allows to train both the backbone and the projector networks through backpropagation without requiring any additional heuristics, such as stop gradient, momentum encoder or clustering operations typically introduced in non-contrastive learning (Chen et al., 2022; He et al., 2020; Grill et al., 2020; Tao et al., 2022; Chen & He, 2021; Tian et al., 2021; Halvagal et al., 2023; Wang et al., 2022). Taken altogether, the *FALCON* projector and loss functions guarantee to train the backbone network in a robust manner, preventing all known forms of collapses. Consequently, this represents the first *failure-proof* non-contrastive SSL solution and we summarize the method together with its PyTorch-like pseudo-code in Figure 2.

## 3.1 MINIMA OF THE LOSS FUNCTION

In the following, we assume that the backbone has infinite capacity. Further discussion about the relaxation of this assumption is left to Appendix B. Therefore, we decouple the study of the objective and its minima from the neural network. In Appendix C, we prove that

**Lemma 1** (Minima). $\forall i \in [n]$ and $j \in [c]$, $\epsilon \leq p_{ij} \leq 1 - \epsilon(c-1)$ and $\epsilon \leq q_{ij} \leq 1 - \epsilon(c-1)$ with $0 \leq \epsilon < 1/c$, then the global minima of the loss function in Eq. 2 jointly satisfy the following conditions:

- ***Invariance*** $\forall i \in [n]$ and $j \in [c]$, $p_{ij} = p'_{ij}$.

- ***Extrema*** $\forall i \in [n]$, $\exists! j \in [c]$, $\forall k \in [c]$ with $k \neq j$, such that $p_{ij} = 1 - \epsilon(c-1)$ and $p_{ik} = \epsilon$. *Here, we refer to the extrema of a probability simplex.*

---

[1] For images $d$ is the product of the width, height and color bands.
[2] $\epsilon = 1e - 8$ throughout the paper.

- **Matched prior** $\forall j \in [c]$, $\frac{1}{n}\sum_{i=1}^{n} p_{ij} = q_j$. Moreover, $\forall j \in [c]$ define $I_{max}(j) \equiv \{i \in [n] : p_{ij} = 1 - \epsilon(c-1)\}$, then $|I_{max}(j)| = \left(\frac{q_j - \epsilon}{1 - c\epsilon}\right) n$.

*The global minimum value of Eq. 2 is bounded by*

$$\mathcal{L}_{FALCON}(\mathcal{D}) \geq -\beta(1 - \epsilon(c-1))\log(1 - \epsilon(c-1)) - \beta\epsilon(c-1)\log\epsilon + H(\boldsymbol{q}) \tag{3}$$

*being equal to $H(\boldsymbol{q})$ whenever $\epsilon = 0$, where $H(\boldsymbol{q})$ is the entropy of $\boldsymbol{q}$.*

The assumptions in the Lemma can always be met by properly choosing arbitrarily small $\epsilon$ to satisfy the relation $0 \leq \epsilon < 1/c$. The results of the Lemma can be intuitively explained by observing that the first two conditions (invariance and extrema) and the last one (matched prior) are mainly a by-product of the invariance and the matching prior losses in Eq. 2, respectively. Indeed, note that the invariance loss can be equivalently expressed as a cross-entropy loss. Therefore, it can be decomposed into the sum of an entropy term for $\boldsymbol{p}_i$ and a Kullback-Leibler (KL) divergence term between $\boldsymbol{p}_i$ and $\boldsymbol{p}_i'$, thus enforcing the extrema condition through the entropy term and the invariance condition through the minimization of the KL one. Minimizing the matching prior loss is equivalent to minimize the KL between $q_j$ and $1/n \sum_{i=1}^{n} p_{ij}$, thus enforcing the matched prior condition. It is also important to specify that while the results of the Lemma are general and valid for any prior distribution $\boldsymbol{q}$, our focus is mainly on the uniform setting. We leave the study of the non-uniform case (et al., 2023) to future work.

An important implication of the Lemma is that the global minima of the *FALCON* objective guarantee to avoid representation and cluster collapses, consequently reducing the need for heuristics, such as stop gradients, momentum encoder and/or specialized clustering layers typically introduced in non-contrastive settings. Indeed, we observe that all data points indexed by $i \in [n]$ are assigned in a hard way to one of the codes available in the dictionary due to the extrema condition. Moreover, the distribution of the assignments follows the result of the matched prior condition, specifically $|I_{max}(j)| = n(q_j - \epsilon)/(1 - c\epsilon)$. For a uniform prior $q_j = 1/c$, we have that $|I_{max}(j)| = n/c$ for all codes $j$, meaning that data points are partitioned and equally assigned to all available codes. Representation collapse is prevented because data points are assigned in a hard fashion to different codes, whereas cluster collapse is avoided because all codes contribute to the partitioning of the data. Moreover, attaining the lower bound value in Eq. 3 gives a certificate for the avoidance of these collapses.

A similar guarantee result has recently appeared in another work (Sansone, 2023), where the same training objective to Eq. 2 is used in addition to a likelihood-based generative term. Their analysis studies each loss term separately, demonstrating their different properties. That is, the generative term prevents representation collapse, the invariance term enforces smoothness and adherence to the cluster assumption, while the matching prior loss prevents cluster collapse. Differently from (Sansone, 2023), we study the objective where the invariance and the matching prior losses are jointly optimized and show through Lemma 1 that they are sufficient to avoid the two collapses without the need of additional terms, like the generative one used in (Sansone, 2023).

## 3.2 PROPERTIES OF THE PROJECTOR

We now turn the analysis to the design of the projector and state the main theorem of this work for the case of $c = f$ (the proof is provided in Appendix D). We will later see how to generalize these results to $c \neq f$. Importantly, the theorem sets the stage for the two key properties of *FALCON*, that is of learning decorrelated and clustered features and avoiding dimensional and intracluster collapses.

**Theorem 1** (Embedding). *Given the projector defined in Eq. 1 with $c = f > 2$ and a dictionary matrix $\boldsymbol{W}$ satisfying the condition $\boldsymbol{W}^T\boldsymbol{W} = f\boldsymbol{I}$, if the optimality conditions of Lemma 1 are met, then the embeddings $\boldsymbol{H}$ satisfy the following relation*

$$\forall i \in [n], \exists! j \in [c] \text{ s.t. } \boldsymbol{h}_i = \alpha_{ij}\boldsymbol{w}_j + \left(\alpha_{ij} - \frac{1}{\sqrt{n}}\right)\sum_{k \neq j}\boldsymbol{w}_k \tag{4}$$

$$\text{with } \alpha_{ij} \in \left\{ \frac{1}{\sqrt{n}}, \left(1 - \frac{2}{c}\right) \frac{1}{\sqrt{n}} \right\}.$$

The theorem tells that at optimality embeddings align with the orthogonal codes from the dictionary. More concretely, each embedding aligns with one code up to some spurious additive term, i.e. the second addend in Eq. 4, whose contribution depends on the admissible values of the coefficient $\alpha_{ij}$ and the remaining codes. Notably, if $\alpha_{ij} = 1/\sqrt{n}$, the spurious term disappears and the embedding shares the same direction of a single code. If $\alpha_{ij} = (1 - 2/c)1/\sqrt{n}$, the contribution of the spurious term becomes non-zero, scaled by a factor of $2/c$. This notion of alignment is important to achieve decorrelated and clustered features, as we will see shortly. The key assumptions to the theorem are the orthogonality of $\boldsymbol{W}$, whose codes define a basis for the embedding space and consequently each embedding can be expressed as a linear combination of the dictionary codes, and normalized embeddings, which allow to constrain the possible values of coefficients for this linear combination. It is important to specify that, for the sake of generality, the theorem considers an orthogonal dictionary matrix, which deviates from the specific choice made in Eq. 1. We will elaborate this detail about $\boldsymbol{W}$ in the next subsection.

The theorem has three important consequences that are distilled in the following corollaries:

**Corollary 1** (Perfect alignment). *Given the assumptions in Theorem 1, if $c \to \infty$, then $\forall i \in [n], \exists! j \in [c]$ we have that $\alpha_{ij} = \frac{1}{\sqrt{n}}$ is unique and $\boldsymbol{h}_i = \frac{1}{\sqrt{n}} \boldsymbol{w}_j$.*

*Proof.* The result in Theorem 1 states that $\forall i \in [n], \exists! j \in [c]$, the coefficients $\alpha_{ij} \in \{1/\sqrt{n}, (1 - 2/c) 1/\sqrt{n}\}$. Taking $c \to \infty$, forces all admissible values of $\alpha_{ij}$ to coincide with a unique value $1/\sqrt{n}$. Substituting this result into Eq. 4 completes the proof. □

This means that the orthogonality of codes, the large size of the dictionary and the normalization of the embeddings are important inductive biases that are sufficient to guarantee perfect alignment to the codes. Indeed, for a large dictionary, each embedding is assigned to only one of the available codes, avoiding spurious terms. We also prove in Appendix E that

**Corollary 2** (Diagonal covariance). *Given Eq. 4 in Theorem 1 and uniform $\boldsymbol{q}$, assume that $\forall i \in [n], \exists! j \in [c]$ such that $\alpha_{ij} = \frac{1}{\sqrt{n}}$, then the covariance of the embeddings is equal to the identity matrix, i.e. $\boldsymbol{H}^T \boldsymbol{H} = \boldsymbol{I}$.*

The assumptions of the corollary can be satisfied by simply choosing a large dictionary along with the other inductive biases discussed for Corollary 1. These biases are sufficient to ensure that the embeddings are decorrelated and span the entire embedding space, thus avoiding dimensional collapse. Moreover, the importance of decorrelated embeddings translates into the property of having both the embedding and representation matrices with full rank. This ensures improved generalization to supervised linear downstream tasks, as shown in Appendix F. Finally, we prove in Appendix G that

**Corollary 3** (Block-diagonal adjacency). *Given Eq. 4 in Theorem 1 and uniform $\boldsymbol{q}$, assume that $\forall i \in [n], \exists! j \in [c]$ such that $\alpha_{ij} = \frac{1}{\sqrt{n}}$, then the adjacency matrix for the embeddings, i.e. $\boldsymbol{H}\boldsymbol{H}^T$, is a block-diagonal matrix with blocks of equal size and their size being equal to $\frac{n}{c}$.*

The orthogonality of the codes and the normalization of the embeddings are therefore sufficient conditions to enforce clustered embeddings. The large size of the dictionary ($c \gg 1$) contributes to decreasing the block size of the adjacency matrix, consequently reducing the effect of intra-cluster collapse.

## 3.3 PRACTICAL CONSIDERATIONS

So far, the analysis has focussed on the case of $c = f$, demonstrating that by choosing large $c$ (also large $f$) leads to decorrelated and clustered embeddings and the prevention of dimensional and intracluster collapses. However, in practice, we rarely have control over the size of the representation $f$. The typical learning setting of SSL takes a backbone network with a fixed $f$ and uses a projector to train it. Therefore, it is natural to ask whether our previous results hold with the increase of $c$ when $f$ is fixed. At a first glance, the answer to this question is negative. Indeed, note that all previous results rely on the assumption of orthogonal $\boldsymbol{W}$, so that the codes span the whole embedding space and also act as an orthogonal basis. Since $f$ is fixed, we can have only $f$ orthogonal codes. We can go beyond

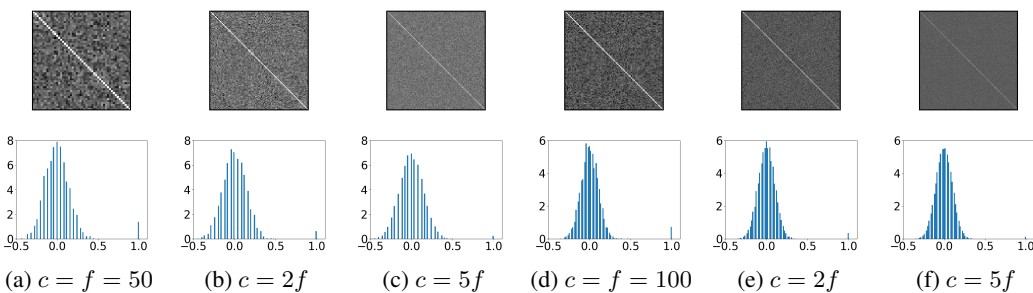

(a) $c = f = 50$  (b) $c = 2f$  (c) $c = 5f$  (d) $c = f = 100$  (e) $c = 2f$  (f) $c = 5f$

Figure 3: **Top:** Illustration of $\boldsymbol{W}^T \boldsymbol{W}$ obtained by randomly sampling $\boldsymbol{W}$. **Bottom:** Normalized histograms of the elements of $\boldsymbol{W}^T \boldsymbol{W}$. Figs. 3a–3c have fixed $f = 50$, whereas Figs. 3d–3f have fixed $f = 100$. $\boldsymbol{W}^T \boldsymbol{W}$ has diagonal values at 1 and random off-diagonal values centered around zero. For larger $c$ and fixed $f$, variance remains constant and quasi-orthogonality is preserved.

such limitation and provide an affirmative answer to the above question by probabilistically relaxing the notion of orthogonal $\boldsymbol{W}$ and leveraging principles from hyperdimensional computing (Kanerva, 2009). More concretely, we can choose codes as in Eq. 1, that is $\boldsymbol{W} \in \{-1, 1\}^{f \times c}$ with elements drawn i.i.d. from a Rademacher distribution, to obtain a quasi-orthogonal dictionary matrix. Indeed, we observe that all columns of $\boldsymbol{W}$ have fixed norm, namely $\|\boldsymbol{w}_j\|_2 = \sqrt{f}$, and that the expected cosine similarity between two codes satisfies the following properties:

$$\mathbb{E}_{\boldsymbol{W}}\{\boldsymbol{w}_j^T \boldsymbol{w}_j\} = \left\{ \begin{array}{ll} 1 & j = j' \\ 0 & j \neq j' \end{array} \right. \quad \text{and } Var_{\boldsymbol{W}}\{\cos(\boldsymbol{w}_j, \boldsymbol{w}_{j'})\} = \frac{1}{f}, \quad \forall j, j' \in [c] \qquad (5)$$

Therefore, the codes are orthogonal to each other on average with the variance being inversely proportional to the size of the representation. In other words, $\boldsymbol{W}^T \boldsymbol{W} = f\boldsymbol{I}$ holds on average independently of the choice of $c$ and all assumptions for Theorem 1 and its corollaries are still satisfied. We illustrate the concept in Figure 3.

As a final practical remark, we introduce a linear and batch normalization layer in Eq. 1 to have well-behaved first-order and second order statistics for the representations throughout training and consequently speedup the convergence of training. Some examples are provided in Appendix A.

## 4 EXPERIMENTS

The experimental analysis is divided into four main parts. Firstly, we compare *FALCON* against non-contrastive approaches from the families of feature decorrelation and cluster-based methods on three image datasets, i.e. SVHN (Netzer et al., 2011), CIFAR-10, CIFAR-100 (Krizhevsky et al., 2009). Secondly, we demonstrate the effects of increasing the dictionary size to validate the results in Corollary 2 and 3 and their implication to generalization on downstream tasks, including clustering and linear probe evaluation. Thirdly, we analyze different families of collapses. Finally, we scale the analysis to ImageNet-100. We use a ResNet-8 backbone network with $f = 128$ for SVHN and CIFAR10, and with $f = 256$ for CIFAR-100, following the methodology from Sansone (2023). For ImageNet-100, we use a standard small ViT with $f = 384$, following the methodology from Caron et al. (2021). The $\beta$ parameter in Eq. 7 is chosen from the range $\{0.01, 0.05, 0.1, 0.25, 0.5, 1, 2.5, 5, 10\}$, so that both terms in the objective are minimized. More details are available in Appendix H). We use the repository from da Costa et al. (2022) for SVHN and CIFAR experiments, and the one from Caron et al. (2021) for ImageNet-100 experiments. Further details are available in Appendices H, K and L.

**Generalization on downstream tasks.** We compare *FALCON* with Barlow Twins (Zbontar et al., 2021), forcing diagonalization of the embedding cross-covariance, SwAV (Caron et al., 2020), using a Sinkhorn-based clustering layer in the projector, Self-Classifier, using a similar loss (Amrani et al., 2022), and GEDI (Sansone & Manhaeve, 2024), using our loss function in conjunction with a multi-layer perceptron projector as in Barlow Twins.[3] We test the downstream performance on clustering using normalized mutual information (NMI) computed between the projector predictions

---

[3]The work in Sansone & Manhaeve (2024) proposes two solutions, one adding a generative term to our objective, named GEDI, and one without, named GEDI *no gen*. Both versions use a standard MLP network for the projector.

Table 1: Test generalization on downstream tasks including clustering and supervised linear probing. Performance are measured in terms of normalized mutual information (NMI), accuracy (Acc.) and are averaged over 5 training runs obtained from random initialization seeds. We test *FALCON* for undercomplete ($c = 10$), complete ($c = f$) and overcomplete ($c = 16384$) dictionaries. For the Self-Classifier, we test the recommended size for the dictionary ($c = k$, with $k$ being the number of ground truth classes) and the overcomplete case ($c = 16384$).

| | **Clustering (NMI)** | | | **Supervised Linear Probing (Acc.)** | | |
|---|---|---|---|---|---|---|
| **Method** | **SVHN** | **CIFAR-10** | **CIFAR-100** | **SVHN** | **CIFAR-10** | **CIFAR-100** |
| Barlow | 0.06±0.02 | 0.05±0.01 | 0.10±0.01 | 0.76±0.01 | 0.65±0.00 | 0.28±0.00 |
| SwAV | 0.03±0.01 | 0.29±0.02 | 0.12±0.07 | 0.45±0.03 | 0.56±0.01 | 0.10±0.06 |
| GEDI *no gen* | 0.07±0.02 | 0.33±0.02 | 0.28±0.00 | 0.63±0.02 | 0.66±0.01 | 0.39±0.00 |
| GEDI | 0.07±0.00 | 0.29±0.01 | 0.25±0.01 | 0.58±0.00 | 0.64±0.01 | 0.38±0.00 |
| Self-Classifier ($c = k$) | 0.07±0.02 | 0.28±0.01 | 0.26±0.00 | 0.58±0.01 | 0.59±0.01 | 0.15±0.00 |
| Self-Classifier ($c = 16384$) | 0.25±0.01 | 0.14±0.10 | 0.17±0.21 | 0.70±0.01 | 0.34±0.18 | 0.16±0.00 |
| *FALCON* (10) | 0.14±0.02 | 0.30±0.01 | 0.16±0.01 | 0.64±0.02 | 0.60±0.01 | 0.15±0.01 |
| *FALCON* ($c = f$) | 0.16±0.00 | 0.26±0.00 | 0.33±0.01 | 0.59±0.01 | 0.60±0.01 | 0.20±0.01 |
| *FALCON* (16384) | **0.31±0.00** | **0.35±0.00** | **0.58±0.00** | **0.78±0.01** | **0.68±0.00** | **0.41±0.00** |

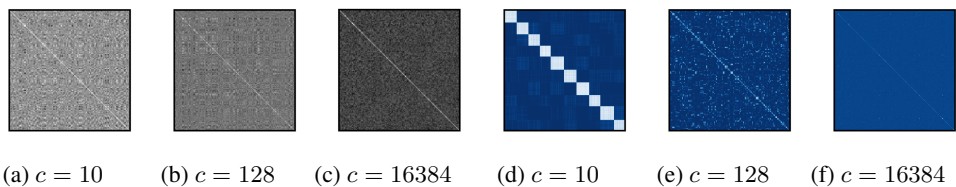

(a) $c = 10$    (b) $c = 128$    (c) $c = 16384$    (d) $c = 10$    (e) $c = 128$    (f) $c = 16384$

Figure 4: Realization of embedding covariance (**left**) and adjacency matrices (**right**) for the whole CIFAR-10 test dataset. Increasing $c$ reduces the value of the off-diagonal elements of the covariance, thus contributing to increase the decorrelation of features (cf. Corollary 2). Moreover, increasing $c$ has the effect to reduce the block sizes of the adjacency matrix (cf. Corollary 3).

and ground truth labels and supervised linear probing on the representations using accuracy (Acc.). In Table 1, we report all results and include three different settings for *FALCON*, each corresponding to the undercomplete ($c<f$), complete ($c=f$) and overcomplete ($c>f$) dictionary case. We observe that Barlow Twins performs well on linear probe evaluation compared to the other baselines, thanks to the connection between feature decorrelation and generalization (cf. Appendix F), whereas SwAV, Self-Classifier and GEDI achieve overall perform better on clustering tasks. We also observe that *FALCON* in the undercomplete case performs comparably well to the other cluster-based baselines despite its difference in the design of the projector. However, the increase of the size of the dictionary in *FALCON* contributes to improve both the clustering and classification performance, as predicted by our theory. Indeed, *FALCON* is the only approach able to systematically exploit this fact.

**Effects of increasing the dictionary size.** We provide additional insights on the benefits of increasing the size of the dictionary. Specifically, we show some examples of embedding covariances and adjacency matrices computed on CIFAR-10 for different values of $c$ in Fig. 4, thus demonstrating that larger dictionary sizes contribute to implicitly diagonalize the covariance as well as to reduce the block sizes in the adjacency, as predicted by our corollaries. More-over, we provide a quantitative evaluation on the downstream tasks in Fig. 5, where we observe a monotonic increase in clustering and classification performance with large values of $c$. Similar results hold for other datasets, cf. Appendix I.

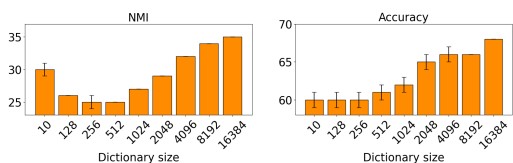

Figure 5: Downstream generalization on CIFAR-10 test dataset, clustering (**left**) and linear evaluation results (**right**).

**Analysis of collapses.** Firstly, we provide some qualitative evidence on the avoidance of representation and cluster collapses, as illustrated in Fig. 6. Indeed, when both losses in the *FALCON* objective are minimized these two failure modes are avoided. Secondly, we investigate dimensional collapse following the methodology proposed in previous work (Jing et al., 2022) by computing the

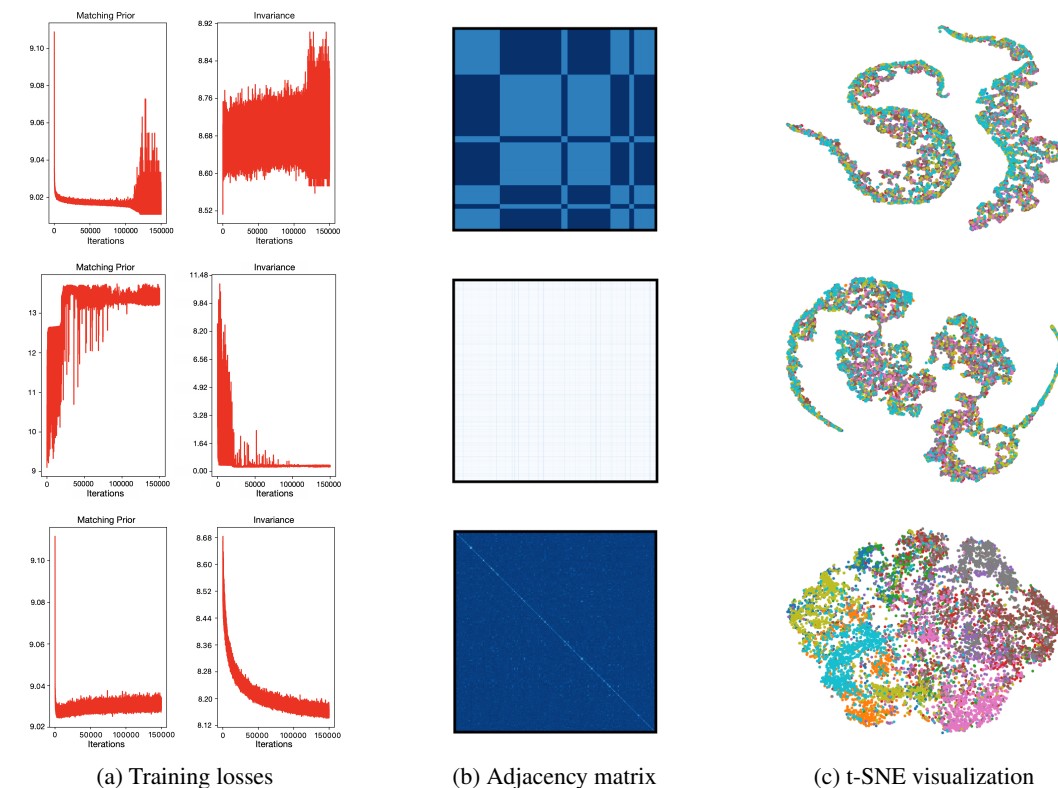

(a) Training losses  (b) Adjacency matrix  (c) t-SNE visualization

Figure 6: Example of bad (**top** with $\beta = 0$ and **middle** with $\beta = 10$) and well-behaved (**bottom**, $\beta = 0.1$) training loss dynamics on CIFAR-10 with dictionary size 8192. When only one term is minimized, the model faces cluster collapse, as demonstrated by the adjacency plots in the top and middle rows (corresponding to 2 and 1 clusters, respectively). However, when both losses are minimized the collapse is avoided. Interestingly, the visualization of the representations reveals the absence of representation collapse in all cases (colors are used to denote different ground truth classes).

Table 2: Supervised linear probing evaluation across different dictionary sizes on backbone trained for 300 epochs on ImageNet-100. Top 1 and Top 5 accuracies are shown.

| Method | Dictionary size | | | | | | | | |
| | 128 | 1024 | 2048 | 4096 | 8192 | 16384 | 32768 | 65536 | 131072 |
|---|---|---|---|---|---|---|---|---|---|
| DINO (Top 1) | 71.8% | 73.6% | 73.9% | 73.6% | 74.3% | 75.0% | 75.1% | 76.2% | 75.8% |
| FALCON (Top 1) | **73.2%** | **74.2%** | **75.0%** | **76.3%** | **76.5%** | **76.9%** | **77.5%** | **78.1%** | **77.1%** |
| DINO (Top 5) | 92.1% | **92.8%** | 92.9% | 92.8% | 93.0% | 93.2% | 93.0% | 94.0% | 94.0% |
| FALCON (Top 5) | **92.2%** | 92.4% | **93.0%** | **93.2%** | **93.7%** | **94.0%** | **94.3%** | 94.2% | **94.5%** |

singular value distribution of the covariance matrix for the embeddings. In Fig. 7a, we observe that for the undercomplete setting only 10 singular values have large values. This is explained by the fact that embeddings align with the 10 codes in the dictionary, as predicted by Theorem 1. When $c$ increases, more and more singular values increase in their value. This provides evidence that the loss function in conjuction with the proposed projector allows to exploit the whole embedding space and avoid dimensional collapse. Results are consistent across all datasets as shown in Appendix J. Finally, we propose to study intracluster collapse by estimating the entropy of the distribution for the representations. We do so by (i) fitting a Gaussian mixture model with diagonal covariance on the representation from the backbone, (ii) estimating the entropy of the distribution through Monte Carlo using $10k$ samples and (iii) repeating the analysis for different number of mixture components, i.e. $\{10, 20, 50, 100, 200, 500, 1000\}$. Intra-cluster collapse is avoided when achieving higher values

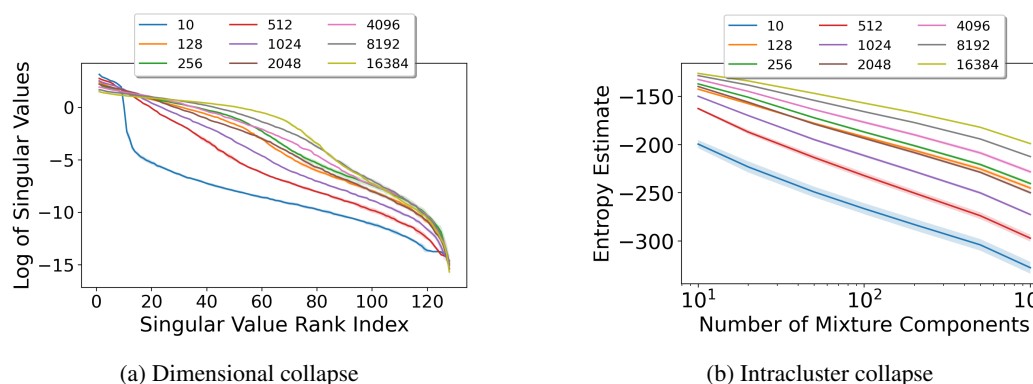

(a) Dimensional collapse           (b) Intracluster collapse

Figure 7: Collapse analysis on CIFAR-10 test data for different dictionary sizes $c$. Results are averaged over 5 training runs obtained from random initialization seeds. **Left:** The singular values of the embedding covariance are in sorted order and logarithmic scale. The curve rises with very large values of $c$, avoiding zero singular values. **Right:** The number of mixture components are in logarithmic scale. The curve rises with very large values of $c$ for all number of mixture components.

of entropy. We illustrate this in Fig. 7b, showcasing improved performance for larger values of dictionary size. Similar behaviour is observed for other datasets, as shown in Appendix J.

**Scaling the analysis to ImageNet-100.** We analyze the effect of the dictionary size on ImageNet-100 using a ViT-small backbone and compare *FALCON* against DINO (Caron et al., 2021). We use the original DINO codebase for the experiments and train the two models for 300 epochs. Table 2 summarizes the results in terms of linear probing evaluation. We observe that *FALCON* is able to make effective and systematic use of larger dictionary sizes, significantly outperforming the performance of DINO. Remarkably, it has been previously observed that DINO can suffer from learning collapsed prototypes (cf. Appendix C in (Garrido et al., 2023a)). This issue is not present in *FALCON* as codes are frozen. This further demonstrates the value of the theoretical guarantees of *FALCON*, translating into a simplified and principled design of the projector and loss function compared to DINO (such as avoiding the use of asymmetric operations like stop gradient, centering operation for the teacher network, use of different temperature parameters for student and teacher networks and exponential moving average update of the teacher parameters).

## 5 CONCLUSIONS

We have distilled the essential principles of non-contrastive self-supervised learning into the design of a projector and loss function that prevent known failure modes, including representation, cluster, dimensional, and intracluster collapses. This approach enables robust training of a backbone network, achieving representations that are both decorrelated and clustered. We have also demonstrated that the resulting solutions improve generalization to supervised downstream tasks when large dictionaries are used in the projector. 'Collapsing' to low-level features like color is a potential failure mode in non-contrastive learning (Chen et al., 2020; Caron et al., 2020; He et al., 2020), often mitigated through data augmentation. Exploring the impact of augmentations on generalization is a promising research direction. Another key focus is improving scalability. The main bottleneck lies in storing and using large dictionaries. Leveraging their bipolar nature, instead of handling them as floating-point collections, could significantly enhance efficiency. Finally, other future directions include scaling the experiments to larger networks and datasets, addressing the non-uniform setting and tackling additional learning and downstream tasks.

## 6 REPRODUCIBILITY STATEMENT

We have detailed all experimental setup in the main paper, with the design of the projector and the loss function. Moreover, we have added all experimental settings in the supplementary material. Code will be released upon publication.

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
