APPENDIX

# A  TRAINING STATISTICS

We provide some examples demonstrating the effectiveness of introducing linear and batch normalization layers in the projector in Fig. 8. This contributes to prevent excessing increase of mean or variance. An alternative and promising approach to the linear and batch normalization layers is to penalize the norm of the representations directly in the objective. We leave this to future investigation.

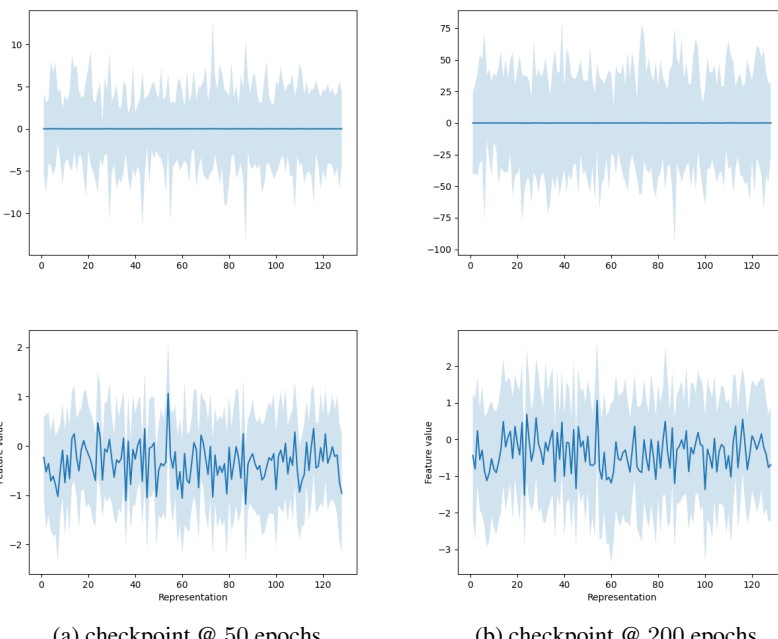

(a) checkpoint @ 50 epochs                    (b) checkpoint @ 200 epochs

Figure 8: Example of mean and standard deviation statistics for the representation features obtained by the backbone network trained on SVHN data. Statistics are computed for a batch of size of 100 samples. Results corresponds to checkpoints for the projector ($c = 4096$) without (**top**) and with (**bottom**) linear and batch normalization layers. Linear and batch normalization layers contribute to stabilize the training by avoiding mean or variance increase.

# B  DISCUSSION ON FINITE CAPACITY

It is important to mention that the global minima for the *FALCON* objective might not be reached when using a backbone network of finite and small capacity. In this case, the avoidance of representation and cluster collapses can still be guaranteed when the invariance and the matching prior losses are both minimized. Indeed, we observe that for representation collapse $p_{ij} = p_j$ for all $i \in [n], j \in [c]$ (i.e. the outputs of the overall network are constant with respect to their inputs) and that the corresponding mimimum value of the objective is given by the following formula

$$\mathcal{L}_{FALCON}(\mathcal{D}) = \beta H(\boldsymbol{p}) + CE(\boldsymbol{q}, \boldsymbol{p})$$

where the first addend arises from the invariance loss, whereas the second one arises from the matching prior one. Notably, the two terms cannot be minimized at the same time due to their competitive nature. For instance, in the case of uniform $\boldsymbol{q}$, the solution of $\boldsymbol{p} = \boldsymbol{q}$ is a minimum for the matching prior loss but not for the invariance one (this is actually a saddle point, as corresponding to the maximum for the entropy term in the above equation).

Cluster collapse occurs whenever $\exists j, k \neq j \in [c]$ such that for all $i \in [n]$, $p_{ij} \leq p_{ik}$. The minimization of the invariance loss forces the whole network to make low entropy predictions,

whereas the minimization of the matching prior loss forces to distribute these predictions across all codes according to $q$. Hence, when both losses are minimized cluster collapse is avoided.

## C  MINIMA OF THE *FALCON* LOSS

*Proof.* We recall here the loss

$$\mathcal{L}_{FALCON}(\mathcal{D}) = -\frac{\beta}{n} \sum_{i=1}^{n} \sum_{j=1}^{c} p_{ij} \log p'_{ij} - \sum_{j=1}^{c} q_j \log \frac{1}{n} \sum_{i=1}^{n} p_{ij}$$

and prove all optimality conditions. Before doing that, we observe that the loss is convex w.r.t. $\mathbf{P}$ when $\mathbf{P}'$ is fixed, as the first addend is a sum of linear terms, whereas the second addend is a sum of convex terms. Similarly, we observe that convexity holds w.r.t. $\mathbf{P}'$ when $\mathbf{P}$ is fixed by exploiting the same reasoning. However, it is important to mention that the loss is not convex globally. This can be shown firstly by computing the Hessian of the first addend w.r.t. both $\mathbf{P}$ and $\mathbf{P}'$ and secondly by observing that the Hessian is not positive semi-definite (we skip the tedious calculation of the Hessian).

**Invariance.** We observe that $\mathbf{P}'$ appears only in the first addend of $\mathcal{L}_{FALCON}$ and that this addend can be equivalently rewritten in the following way:

$$-\frac{\beta}{n} \sum_{i=1}^{n} \sum_{j=1}^{c} p_{ij} \log p'_{ij} = -\frac{\beta}{n} \sum_{i=1}^{n} \sum_{j=1}^{c} p_{ij} \log p_{ij} - \frac{\beta}{n} \sum_{i=1}^{n} \sum_{j=1}^{c} p_{ij} \log \frac{p'_{ij}}{p_{ij}}$$

$$= \frac{\beta}{n} \sum_{i=1}^{n} H(\mathbf{p}_i) + \frac{\beta}{n} \sum_{i=1}^{n} KL(\mathbf{p}_i \| \mathbf{p}'_i) \tag{6}$$

where $H(.), KL(.)$ are the entropy and Kullback-Leibler divergence, respectively. Therefore minimizing $\mathcal{L}_{FALCON}$ w.r.t. $\mathbf{P}'$ is equivalent to minimizing Eq. 6. The solution is given by $\mathbf{p}_i = \mathbf{p}'_i$, $\forall i \in [n]$, thus proving the invariance condition.

**Extrema.** We first leverage the invariance condition, $\mathbf{p}_i = \mathbf{p}'_i$, $\forall i \in [n]$, and rewrite $\mathcal{L}_{FALCON}$ accordingly:

$$\boxed{\mathcal{L}_{FALCON}(\mathcal{D}) = -\frac{\beta}{n} \sum_{i=1}^{n} \sum_{j=1}^{c} p_{ij} \log p_{ij} - \sum_{j=1}^{c} q_j \log \frac{1}{n} \sum_{i=1}^{n} p_{ij}} \tag{7}$$

We observe that the loss in Eq. 7 is convex w.r.t. $\mathbf{P}$. Therefore, we can obtain its optimality conditions, by deriving the closed-form solutions for the minima of the second addend in Eq. 7, and then constraining the optimization of the first addend with these solutions and deriving the corresponding minima.

Let's start by considering the following constrained convex minimization problem, obtained from the first addend in Eq. 7, with $n, \beta$ being dropped as being constant for the optimization:

$$\min_{\mathbf{P}} -\sum_{i=1}^{n} \sum_{j=1}^{c} p_{ij} \log p_{ij}$$

$$\text{s.t.} \quad \sum_{j=1}^{c} p_{ij} = 1, \quad \forall i \in [n]$$

$$\epsilon \le p_{ij} \le 1 - \epsilon(c - 1), \quad \forall i \in [n], j \in [c],$$

$$\tag{8}$$

and the corresponding Lagrangian with multipliers $\mathbf{\Lambda}, \mathbf{\Delta} \in \mathbb{R}_+^{n \times c}, \boldsymbol{\nu} \in \mathbb{R}^n$ is:

$$\mathcal{L}_1(\mathbf{P}; \mathbf{\Lambda}, \mathbf{\Delta}, \mathbf{\Omega}, \boldsymbol{\nu}) \equiv -\sum_{i=1}^{n} \sum_{j=1}^{c} p_{ij} \log p_{ij} + \sum_{i=1}^{n} \nu_i \left( \sum_{j=1}^{c} p_{ij} - 1 \right) +$$

$$+ \sum_{i=1}^{n} \sum_{j=1}^{c} \left[ \lambda_{ij}(\epsilon - p_{ij}) + \delta_{ij}(p_{ij} - 1 + \epsilon(c-1)) \right] \tag{9}$$

We observe that the Lagrangian is constructed so as to satisfy the following relation

$$- \sum_{i=1}^{n} \sum_{j=1}^{c} p_{ij} \log p_{ij} \geq \mathcal{L}_1(\mathbf{P}; \mathbf{\Lambda}, \mathbf{\Delta}, \mathbf{\Omega}, \boldsymbol{\nu}) \tag{10}$$

Let's maximize $\mathcal{L}_1$ w.r.t. $\mathbf{P}$ by setting $\nabla_{p_{ij}} \mathcal{L}_1 = 0$. This leads to the following closed-form expression:

$$p_{ij}^* = e^{-1 - \lambda_{ij} + \nu_i + \delta_{ij}} \quad \forall i \in [n], j \in [c] \tag{11}$$

By evaluating $\mathcal{L}_1$ at the solutions in Eq. 11, we obtain the Lagrange dual function

$$\mathcal{L}_1(\mathbf{P}^*; \mathbf{\Lambda}, \mathbf{\Delta}, \mathbf{\Omega}, \boldsymbol{\nu}) = n + \sum_{i=1}^{n} \left\{ -\nu_i + \sum_{j=1}^{c} \left[ \lambda_{ij}\epsilon - \delta_{ij}(1 - \epsilon(c-1)) \right] \right\} \tag{12}$$

The Lagrange multipliers in Eq. 12 depend on the values of $\mathbf{P}^*$ through the Karush-Kuhn-Tucker (KKT) conditions. We distinguish two main cases for $\mathbf{P}^*$, each leading to different evaluation of the Lagrange dual function:

- *Case 1.* When all probability values touch their extrema, such as

$$\forall i \in [n], \exists! j \in [c], \forall k \in [c] \text{ with } k \neq j \text{ s.t. } p_{ij}^* = 1 - \epsilon(c-1) \text{ and } p_{ik}^* = \epsilon$$

  By the KKT conditions (i.e. complementary slackness), we have that $\lambda_{ij} = 0$ and $\delta_{ik} = 0$, whereas $\lambda_{ik} \geq 0, \delta_{ij} \geq 0$. By substituting these conditions in Eq. 12, we obtain that

$$\mathcal{L}_1(\mathbf{P}^*; \mathbf{\Lambda}, \mathbf{\Delta}, \mathbf{\Omega}, \boldsymbol{\nu})|_{\{\lambda_{ij} = \delta_{ik} = 0\}} = n + \sum_{i=1}^{n} \left\{ -\nu_i - \delta_{ij}(1 - \epsilon(c-1)) + \sum_{k \neq j} \lambda_{ik}\epsilon \right\} \tag{13}$$

  By taking into account also Eq. 11, we have that $\forall i \in [n], \exists! j \in [c], \forall k \in [c]$

$$\delta_{ij} = 1 - \nu_i + \log(1 - \epsilon(c-1)) \text{ and } \lambda_{ik} = -1 + \nu_i - \log \epsilon \tag{14}$$

  And by substituting Eq. 14 into Eq. 13, we obtain that

$$\mathcal{L}_1(\mathbf{P}^*; \mathbf{\Lambda}, \mathbf{\Delta}, \mathbf{\Omega}, \boldsymbol{\nu})|_{\{\lambda_{ij} = \delta_{ik} = 0\} \text{ and Eq. } 14} = -n(1 - \epsilon(c-1)) \log(1 - \epsilon(c-1)) - \\ - n\epsilon(c-1) \log \epsilon \tag{15}$$

- *Case 2.* When all probability values never touch the highest extrema, such as

$$\forall i \in [n], j \in [c], \text{ s.t. } p_{ij}^* < 1 - \epsilon(c-1)$$

  By KKT conditions, we have that $\delta_{ij} = 0$. By substituting these conditions in Eq. 12, we obtain that

$$\mathcal{L}_1(\mathbf{P}^*; \mathbf{\Lambda}, \mathbf{\Delta}, \mathbf{\Omega}, \boldsymbol{\nu})|_{\{\delta_{ij} = 0\}} = n + \sum_{i=1}^{n} \left\{ -\nu_i + \sum_{j=1}^{c} \lambda_{ij}\epsilon \right\} \tag{16}$$

  which always satisfies the inequality

$$\mathcal{L}_1(\mathbf{P}^*; \mathbf{\Lambda}, \mathbf{\Delta}, \mathbf{\Omega}, \boldsymbol{\nu})|_{\{\delta_{ij} = 0\}} \geq \mathcal{L}_1(\mathbf{P}^*; \mathbf{\Lambda}, \mathbf{\Delta}, \mathbf{\Omega}, \boldsymbol{\nu})|_{\{\lambda_{ij} = \delta_{ik} = 0\}} \tag{17}$$

  and therefore also

$$\mathcal{L}_1(\mathbf{P}^*; \mathbf{\Lambda}, \mathbf{\Delta}, \mathbf{\Omega}, \boldsymbol{\nu})|_{\{\delta_{ij} = 0\}} \geq \mathcal{L}_1(\mathbf{P}^*; \mathbf{\Lambda}, \mathbf{\Delta}, \mathbf{\Omega}, \boldsymbol{\nu})|_{\{\lambda_{ij} = \delta_{ik} = 0\} \text{ and Eq. } 14} \tag{18}$$

Finally, we observe that the objective of the optimization problem of Eq. 8 evaluated at the solutions of *Case 1* is

$$-\sum_{i=1}^{n}\sum_{j=1}^{c} p_{ij} \log p_{ij} = \mathcal{L}_1(\mathbf{P}^*; \mathbf{\Lambda}, \mathbf{\Delta}, \mathbf{\Omega}, \boldsymbol{\nu})|_{\{\lambda_{ij}=\delta_{ik}=0\} \text{ and Eq. } 14} \quad (19)$$

And by leveraging also the result in Eq. 18, we can state that the solutions of *Case 1* are the global minima of the objective in Eq. 8. Thus concluding the proof for the extrema condition.

**Matched prior.** We consider the minimization of the second addend in Eq. 7 subject to the extrema condition

$$\min_{\mathbf{P}} -\sum_{j=1}^{c} q_j \log \frac{1}{n}\sum_{i=1}^{n} p_{ij}$$

$$\text{s.t.} \quad \sum_{j=1}^{c} p_{ij} = 1, \quad \forall i \in [n]$$

$$p_{ij} \in \{\epsilon, 1 - \epsilon(c-1)\}, \quad \forall i \in [n], j \in [c], \quad (20)$$

Let's define $\tilde{p}_j \equiv \frac{1}{n}\sum_{i=1}^{n} p_{ij}$ for all $j \in [c]$ and observe that $\sum_{j=1}^{c} \tilde{p}_j = 1$ and $\epsilon \leq \tilde{p}_j \leq 1 - \epsilon(c-1)$. Therefore, we can rewrite the problem in Eq. 20 equivalently

$$\min_{\mathbf{P}} -\sum_{j=1}^{c} q_j \log \tilde{p}_j$$

$$\text{s.t.} \quad \sum_{j=1}^{c} \tilde{p}_j = 1,$$

$$\epsilon \leq \tilde{p}_j \leq 1 - \epsilon(c-1), \quad \forall j \in [c], \quad (21)$$

Now, we observe that the optimization objective satisfies the following equality

$$-\sum_{j=1}^{c} q_j \log \tilde{p}_j = H(\boldsymbol{q}) + KL(\mathbf{q}\|\tilde{\boldsymbol{p}}) \quad (22)$$

The minimum for Eq. 22 is obtained at $\mathbf{q} = \tilde{\boldsymbol{p}}$ and this solution satisfies the constraints in Eq. 21 because $\epsilon \leq q_j \leq 1 - \epsilon(c-1)$ for all $j \in [c]$ (indeed we can always choose $\epsilon$ to satisfy the inequality), thus being the global optimum. In other words, we have that $\frac{1}{n}\sum_{i=1}^{n} p_{ij} = q_j$ for all $j \in [c]$.

Finally, recall that $I_{max}(j) \equiv \{i \in [n] : p_{ij} = 1 - \epsilon(c-1)\}, \forall j \in [c]$, which identifies all elements having the highest possible value of probability in $\boldsymbol{P}$. We observe that

$$\sum_{i=1}^{n} p_{ij} = \sum_{i \in I_{max}(j)} p_{ij} + \sum_{i \notin I_{max}(j)} p_{ij}$$

$$= \sum_{i \in I_{max}(j)} (1 - \epsilon(c-1)) + \sum_{i \notin I_{max}(j)} \epsilon \quad \text{(by Extrema condition)}$$

$$= |I_{max}(j)|(1 - c\epsilon) + n\epsilon$$

By the condition $\frac{1}{n}\sum_{i=1}^{n} p_{ij} = q_j$ and the above relation we have that

$$|I_{max}(j)|(1 - c\epsilon) + n\epsilon = nq_j, \quad \forall j \in [c]$$

or equivalently that

$$|I_{max}(j)| = \left(\frac{q_j - \epsilon}{1 - c\epsilon}\right) n \quad (23)$$

Now, for the case of uniform prior, Eq. 23 becomes

$$q_j = \frac{1}{c} \implies |I_{max}(j)| = \frac{n}{c}, \quad \forall j \in [c] \tag{24}$$

This concludes the proof for the matching prior condition.

Finally the global minimum value of the *FALCON* objective can be obtained by dividing Eq. 15 by $n$ and adding the entropy term (as for the result obtained by the matched prior condition). This concludes the proof of the Lemma. □

# D  EMBEDDING THEOREM

*Proof.* Recall the extrema condition from Lemma 1, that is

$$\forall i \in [n], \exists! j \in [c], \forall k \in [c] \text{ with } k \neq j \text{ s.t. } p_{ij} = 1 - \epsilon(c - 1) \text{ and } p_{ij} = \epsilon$$

Moreover, due to orthogonality of $W$ we can express the *Span* condition, i.e. $h_i = \sum_{j'=1}^{c} \alpha_{ij'} w_{j'}$ for all $i \in [n]$ with $\alpha_{ij} \in \mathbb{R}$, This fact leads us to the following equation

$$p_{ij} = \frac{e^{w_j^T h_i / \tau}}{\sum_{j''=1}^{c} e^{w_{j''}^T h_i / \tau}} \underbrace{=}_{Span} \frac{e^{\alpha_{ij} f / \tau}}{\sum_{j'=1}^{c} e^{\alpha_{ij'} f / \tau}} \quad \forall i \in [n], j \in [c] \tag{25}$$

Combining the extrema condition with Eq. 25 gives us a system of equations for each $i \in [n]$

$$\begin{cases} \frac{e^{\alpha_{ij} f / \tau}}{\sum_{j'=1}^{c} e^{\alpha_{ij'} f / \tau}} & = 1 - \epsilon(c - 1) \\ \frac{e^{\alpha_{ik} f / \tau}}{\sum_{j'=1}^{c} e^{\alpha_{ij'} f / \tau}} & = \epsilon & \forall k \neq j \end{cases}$$

By taking the logarithm on both sides of the two equations and resolving the above system, the solution is equal to

$$\alpha_{ik} = \alpha_{ij} - \frac{\tau}{f} \log\left(\frac{1 - \epsilon(c - 1)}{\epsilon}\right)$$

$$= \alpha_{ij} - \frac{1}{\sqrt{n}} \quad \forall k \neq j \tag{26}$$

where the last equality holds due to the choice $\tau = f / (\sqrt{n} \log((1 - \epsilon(c - 1))/\epsilon))$. Using Eq. 26 in the *Span* condition gives us the following result

$$\forall i \in [n], \exists! j \in [c] \text{ s.t. } h_i = \alpha_{ij} w_j + \left(\alpha_{ij} - \frac{1}{\sqrt{n}}\right) \sum_{k \neq j} w_k \tag{27}$$

Note that the $\alpha_{ij}$ could potentially take any value in $\alpha_{ij} \in \mathbb{R}$. This is not allowed as embeddings are normalized by design choice (cf. Eq. 1). Indeed, the norm of the embeddings can be rewritten to exploit Eq. 27

$$\|h_i\|_2^2 = h_i^T h_i$$

$$\underbrace{=}_{\text{Eq. 27}} c f \alpha_{ij}^2 - \frac{2(c - 1)f}{\sqrt{n}} \alpha_{ij} + \frac{f}{n}(c - 1) \tag{28}$$

and by equating Eq. 28 to the fact that embeddings are normalized $\|h_i\|_2^2 = \frac{f}{n}$ for all $i \in [n]$ we obtain the following quadratic equation

$$\alpha_{ij}^2 - \frac{2(c - 1)f}{\sqrt{n}} \alpha_{ij} + \frac{f}{n}(c - 1) - \frac{f}{n} = 0$$

whose solutions are given by

$$\alpha_{ij} = \begin{cases} \frac{1}{\sqrt{n}} \\ \left(1 - \frac{2}{c}\right) \frac{1}{\sqrt{n}} \end{cases}$$

This concludes the proof. □

## E  DIAGONAL COVARIANCE

*Proof.* Recall from Theorem 1 that

$$\forall i \in [n], \exists! j \in [c] \text{ s.t. } \boldsymbol{h}_i = \alpha_{ij}\boldsymbol{w}_j + \left(\alpha_{ij} - \frac{1}{\sqrt{n}}\right)\sum_{k \neq j}\boldsymbol{w}_k$$

By assumption $\alpha_{ij} = \frac{1}{\sqrt{n}}$ and therefore

$$\forall i \in [n], \exists! j \in [c] \text{ s.t. } \boldsymbol{h}_i = \frac{1}{\sqrt{n}}\boldsymbol{w}_j \tag{29}$$

meaning that the rows of $\boldsymbol{H}$ are equal up to a constant to the codes in the dictionary and that they span the same space of the columns of $\boldsymbol{W}$, namely the whole embedding space. We can therefore express $\boldsymbol{H}$ as linear combination of $\boldsymbol{W}$.

Without loss of generality, we can always define $\boldsymbol{H}$ so as to ensure that nearby rows are associated to the same codes in the dictionary. Therefore, by combining this with Eq. 29 we have that

$$\boldsymbol{H} = \boldsymbol{A}^T\boldsymbol{W}^T$$

with

$$\boldsymbol{A} = \begin{pmatrix} \frac{1}{\sqrt{n}}\mathbf{1}_{n/c}^T & \mathbf{0} & \cdots & \mathbf{0} \\ \mathbf{0} & \frac{1}{\sqrt{n}}\mathbf{1}_{n/c}^T & \cdots & \mathbf{0} \\ \vdots & \vdots & \ddots & \vdots \\ \mathbf{0} & \mathbf{0} & \cdots & \frac{1}{\sqrt{n}}\mathbf{1}_{n/c}^T \end{pmatrix} \in \mathbb{R}^{c \times n}$$

where $\mathbf{1}_{n/c}$ is a vector containing $n/c$ ones (whose size follows due to the assumption on uniformity of $\boldsymbol{q}$). Importantly, matrix $\boldsymbol{A}$ satisfies the following property

$$\boldsymbol{A}\boldsymbol{A}^T = \frac{1}{c}\boldsymbol{I} \tag{30}$$

Therefore, we have that

$$\boldsymbol{H}^T\boldsymbol{H} = \boldsymbol{W}\boldsymbol{A}\boldsymbol{A}^T\boldsymbol{W}^T$$

$$\underbrace{=}_{\text{Eq. 30}} \frac{1}{c}\boldsymbol{W}\boldsymbol{W}^T$$

$$= \boldsymbol{I}$$

where the last equality simply follows by the orthogonality condition $\boldsymbol{W}^T\boldsymbol{W} = f\boldsymbol{I}$ and the fact that $\boldsymbol{W}$ is a square matrix ($c = f$). Indeed, we have that

$$\boldsymbol{W}^T\boldsymbol{W} = f\boldsymbol{I}$$

$$\boldsymbol{W}\boldsymbol{W}^T\boldsymbol{W} = f\boldsymbol{I}\boldsymbol{W}$$

$$(\boldsymbol{W}\boldsymbol{W}^T)\boldsymbol{W} = (f\boldsymbol{I})\boldsymbol{W}$$

$$\boldsymbol{W}\boldsymbol{W}^T = f\boldsymbol{I}$$

thus concluding the proof. □

## F  GENERALIZATION TO SUPERVISED LINEAR DOWNSTREAM TASK

We first observe that by the results of Theorem 1 and the uniformity of $\boldsymbol{q}$, $\boldsymbol{H}$ has full rank. Moreover, considering that $\boldsymbol{H}$ is a function of $\boldsymbol{Z}$ through the first layer of the projector in Eq. 1, $\boldsymbol{Z}$ must be also full rank. As a consequence,

$$\boldsymbol{Z}^T\boldsymbol{Z} \text{ has full rank.} \quad \textit{(Full Rank Property)} \tag{31}$$

Now, we recall an existing result for generalization to supervised downstream tasks from Shwartz-Ziv et al. (2023) (Section 6.1) and demonstrate that the *Full Rank Property* reduces the generalization error.

Indeed, consider a classification problem with $r$ classes. Given an unlabeled dataset $\mathcal{D}$, used for training *FALCON*, with the corresponding unknown ground truth labels $\boldsymbol{Y}_{\mathcal{D}} \in \mathbb{R}^{n \times r}$ and a supervised dataset $\mathcal{S} = \{(\boldsymbol{x}_i, \boldsymbol{y}_i)\}_{i=1}^{m}$, with $\boldsymbol{y}_i$ being the rows of the label matrix $\boldsymbol{Y}_{\mathcal{S}} \in \mathbb{R}^{m \times r}$, define $\boldsymbol{Z} \in \mathbb{R}^{n \times f}$ and $\bar{\boldsymbol{Z}} \in \mathbb{R}^{m \times f}$ the representations obtained by feeding datasets $\mathcal{D}$ and $\mathcal{S}$, respectively, through the backbone network $g$. Moreover, define

$$\boldsymbol{P}_{\mathcal{D}} \equiv \boldsymbol{I} - \boldsymbol{Z}(\boldsymbol{Z}^T \boldsymbol{Z})^{\dagger} \boldsymbol{Z}^T$$
$$\boldsymbol{P}_{\mathcal{S}} \equiv \boldsymbol{I} - \bar{\boldsymbol{Z}}(\bar{\boldsymbol{Z}}^T \bar{\boldsymbol{Z}})^{\dagger} \bar{\boldsymbol{Z}}^T$$

where symbol $\cdot^{\dagger}$ denotes the pseudo-inverse. Now, suppose we train a linear classifier with parameters $\boldsymbol{U} \in \mathbb{R}^{f \times r}$ on the latent representations obtained from dataset $\mathcal{S}$ through the following supervised loss

$$\ell_{\boldsymbol{x}, \boldsymbol{y}}(\boldsymbol{U}) \equiv \|g(\boldsymbol{x})\boldsymbol{U} - \boldsymbol{y}\|_2^2 + \|\boldsymbol{U}\|_F$$

Then, we can state the following theorem

**Th. 1** (restated from Shwartz-Ziv et al. (2023)). $\forall \delta > 0$ *with probability at least* $1 - \delta$, *we have that*

$$\mathbb{E}_{\boldsymbol{x}, \boldsymbol{y}}\{\ell_{\boldsymbol{x}, \boldsymbol{y}}(\boldsymbol{U})\} \leq \frac{1}{n} \sum_{i=1}^{n} \|g(\boldsymbol{x}_i) - g(\boldsymbol{x}_i')\|_2 + \frac{2}{m} \mathbb{E}_{\mathcal{D}, \boldsymbol{\xi}} \left\{ \sup_g \sum_{i=1}^{n} \xi_i \|g(\boldsymbol{x}_i) - g(\boldsymbol{x}_i')\|_2 \right\} +$$
$$+ \frac{2}{\sqrt{n}} \|\boldsymbol{P}_{\mathcal{D}} \boldsymbol{Y}_{\mathcal{D}}\|_F + \frac{1}{\sqrt{m}} \|\boldsymbol{P}_{\mathcal{S}} \boldsymbol{Y}_{\mathcal{S}}\|_F + const(n, m) \tag{32}$$

*where $\boldsymbol{\xi}$ is a vector of i.i.d. Rademacher random variables.*

Therefore, the expected supervised loss in Eq. 32 can be reduced by minimizing its upper bound. Note that the first addend in Eq. 32 is minimized by the *FALCON* loss, whereas the second addend is also statistically minimized when $n$ is large. The third addend refers to the contribution term for the classification on the unlabeled data. While ground truth $\boldsymbol{Y}_{\mathcal{D}}$ is unknown, this addend can be minimized by exploiting the following relation

$$\|\boldsymbol{P}_{\mathcal{D}} \boldsymbol{Y}_{\mathcal{D}}\|_F \leq \|\boldsymbol{P}_{\mathcal{D}}\|_F \|\boldsymbol{Y}_{\mathcal{D}}\|_F$$

Indeed, note that in order to minimize the left-hand side of the inequality, it suffices to minimize the term $\|\boldsymbol{P}_{\mathcal{D}}\|_F$, which occurs when $\boldsymbol{Z}^T \boldsymbol{Z}$ has maximum rank. This is our case due to the *Full Rank Property*. Finally, by the same argument used for the third term in Eq. 32, we can minimize the fourth one by having $\bar{\boldsymbol{Z}}^T \bar{\boldsymbol{Z}}$ with maximum rank. This condition holds because $\boldsymbol{Z}^T \boldsymbol{Z}$ and $\bar{\boldsymbol{Z}}^T \bar{\boldsymbol{Z}}$ concentrate to each other by concentration inequalitites (cf. Shwartz-Ziv et al. (2023) for more details).

To summarize, minimizing the *FALCON* loss ensures that we reduce the invariance of representations to data augmentations and increase the rank of the representation covariance. This leads to a decrease of the generalization error as from the result of Theorem 1.

# G  BLOCK-DIAGONAL ADJACENCY

*Proof.* The proof follows step by step the one for the diagonal covariance except for the fact that

$$\boldsymbol{H}\boldsymbol{H}^T = \boldsymbol{A}^T \boldsymbol{W}^T \boldsymbol{W} \boldsymbol{A} \underbrace{=}_{\boldsymbol{W}^T \boldsymbol{W} = f\boldsymbol{I}} f\boldsymbol{A}^T \boldsymbol{A} = f\boldsymbol{B}_{\boldsymbol{A}}$$

where

$$\boldsymbol{B}_{\boldsymbol{A}} \equiv \boldsymbol{A}^T \boldsymbol{A} = \begin{pmatrix} \boldsymbol{1}_{\frac{n}{c} \times \frac{n}{c}} & \boldsymbol{0} & \cdots & \boldsymbol{0} \\ \boldsymbol{0} & \frac{1}{n} \boldsymbol{1}_{\frac{n}{c} \times \frac{n}{c}} & \cdots & \boldsymbol{0} \\ \vdots & \vdots & \ddots & \vdots \\ \boldsymbol{0} & \boldsymbol{0} & \cdots & \frac{1}{n} \boldsymbol{1}_{\frac{n}{c} \times \frac{n}{c}} \end{pmatrix} \in \mathbb{R}^{n \times n}$$

and $\boldsymbol{1}_{\frac{n}{c} \times \frac{n}{c}}$ is a matrix of ones. This concludes the proof. $\qquad \square$

Table 3: Resnet architecture. Conv2D(A,B,C) applies a 2d convolution to input with B channels and produces an output with C channels using stride (1, 1), padding (1, 1) and kernel size (A, A).

| Name | Layer | Res. Layer |
|---|---|---|
| Block 1 | Conv2D(3,3,F) LeakyRELU(0.2) Conv2D(3,F,F) AvgPool2D(2) | AvgPool2D(2) Conv2D(1,3,F) no padding |
| | Sum | |
| Block 2 | LeakyRELU(0.2) Conv2D(3,F,F) LeakyRELU(0.2) Conv2D(3,F,F) AvgPool2D(2) | |
| Block 3 | LeakyRELU(0.2) Conv2D(3,F,F) LeakyRELU(0.2) Conv2D(3,F,F) | |
| Block 4 | LeakyRELU(0.2) Conv2D(3,F,F) LeakyRELU(0.2) Conv2D(3,F,F) AvgPool2D(all) | |

Table 4: Hyperparameters (in terms of optimizer and data augmentation) used in SVHN, CIFAR-10 and CIFAR-100 experiments.

| Class | Name param. | SVHN | CIFAR-10 | CIFAR-100 |
|---|---|---|---|---|
| Data augment. | Color jitter prob. | 0.1 | 0.1 | 0.1 |
| | Gray scale prob. | 0.1 | 0.1 | 0.1 |
| | Random crop | Yes | Yes | Yes |
| | Additive Gauss. noise (std) | 0.03 | 0.03 | 0.03 |
| | Random horizontal flip | No | Yes | Yes |
| Optimizer | Batch size | 64 | 64 | 64 |
| | Epochs | 20 | 200 | 200 |
| | Adam $\beta_1$ | 0.9 | 0.9 | 0.9 |
| | Adam $\beta_2$ | 0.999 | 0.999 | 0.999 |
| | Learning rate | $1e-4$ | $1e-4$ | $1e-4$ |

## H EXPERIMENTAL DETAILS ON SVHN, CIFAR10 AND CIFAR100

**Training.** We used a ResNet-8 (details are provided in Table 3. We consider the hyperparameters in Table 4 for training. Beta is chosen to ensure both losses are minimized, cf. Table 5.

**Evaluation.** For linear probe evaluation, we followed standard practice by removing the projector head and train a linear classifier on the backbone representation. We train the classifier with Adam optimizer for 100 epochs and learning rate equal to $1e-2$.

## I ADDITIONAL RESULTS ON DICTIONARY SIZE

We provide additional visualization results for the covariance and adjacency matrices on SVHN and CIFAR-10, cf. Figs. 9, 10. Moreover, we add the analysis of generalization on downstream tasks on SVHN and CIFAR-100 varying the size of the dictionary in Figs 11, 12.

Table 5: Values of $\beta$ hyperparameter. This is chosen from the range $\{0.01, 0.05, 0.1, 0.25, 0.5, 1, 2.5, 5, 10\}$ to ensure that both losses are minimized.

| Dictionary Size | 10 | 128 | 256 | 512 | 1024 | 2048 | 4096 | 8192 | 16384 |
|---|---|---|---|---|---|---|---|---|---|
| SVHN | 0.5 | 0.5 | 0.5 | 0.25 | 0.1 | 0.1 | 0.1 | 0.1 | 0.1 |
| CIFAR-10 | 0.5 | 0.5 | 0.5 | 0.25 | 0.1 | 0.1 | 0.1 | 0.1 | 0.1 |
| CIFAR-100 | 0.5 | 0.5 | 0.5 | 0.25 | 0.1 | 0.1 | 0.1 | 0.1 | 0.1 |

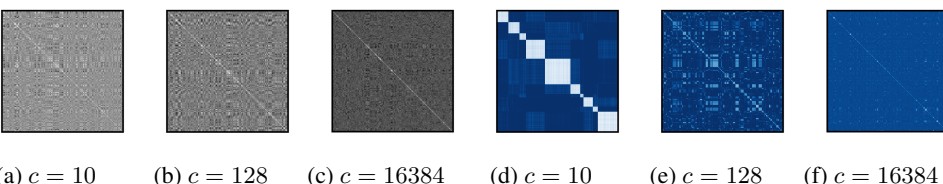

(a) $c = 10$    (b) $c = 128$    (c) $c = 16384$    (d) $c = 10$    (e) $c = 128$    (f) $c = 16384$

Figure 9: Realization of embedding covariance (**left**) and adjacency matrices (**right**) for the whole SVHN test dataset. Increasing $c$ reduces the value of the off-diagonal elements of the covariance, thus contributing to increase the decorrelation of features (cf. Corollary 2). Moreover, increasing $c$ has the effect to reduce the block sizes of the adjacency matrix (cf. Corollary 3).

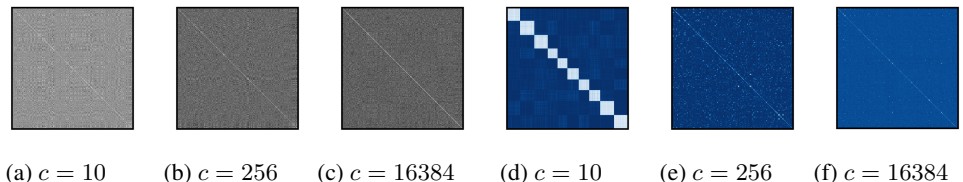

(a) $c = 10$    (b) $c = 256$    (c) $c = 16384$    (d) $c = 10$    (e) $c = 256$    (f) $c = 16384$

Figure 10: Realization of embedding covariance (**left**) and adjacency matrices (**right**) for the whole CIFAR-100 test dataset. Increasing $c$ reduces the value of the off-diagonal elements of the covariance, thus contributing to increase the decorrelation of features (cf. Corollary 2). Moreover, increasing $c$ has the effect to reduce the block sizes of the adjacency matrix (cf. Corollary 3).

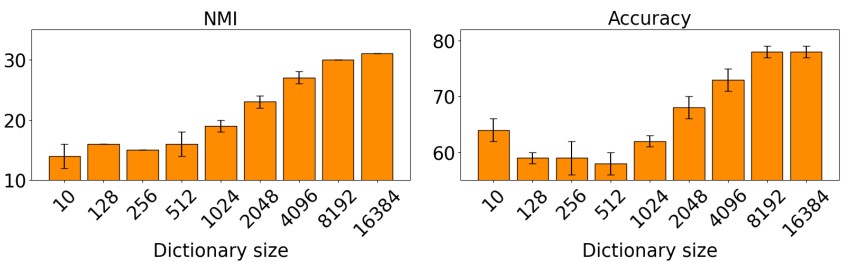

Figure 11: Analysis of downstream generalization for different values of dictionary size on SVHN dataset.

## J ADDITIONAL ANALYSIS ON COLLAPSES

We provide additional results for the collapses on SVHN and CIFAR100. Specifically, in Fig. 13 we show the analysis of dimensional collapses, whereas in Fig. 14 we show the one for intracluster collapse.

## K EXPERIMENTAL DETAILS ON IMAGENET-100

**Training.** We used a ViT-small backbone network and train it for 100 epochs with learning rate equal to $5e - 4$ and batch-size per GPU equal to $64$ on a node with 8 NVIDIA A100 GPUs. Beta is selected

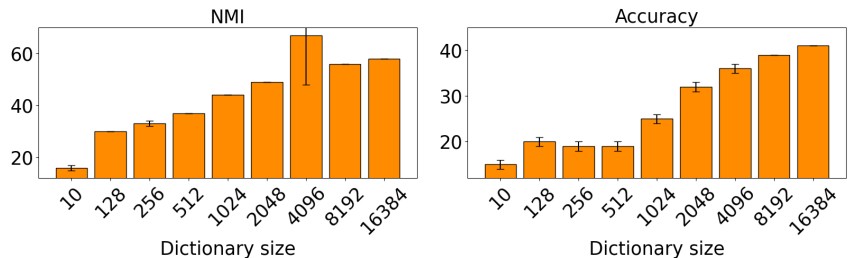

Figure 12: Analysis of downstream generalization for different values of dictionary size on CIFAR-100 dataset.

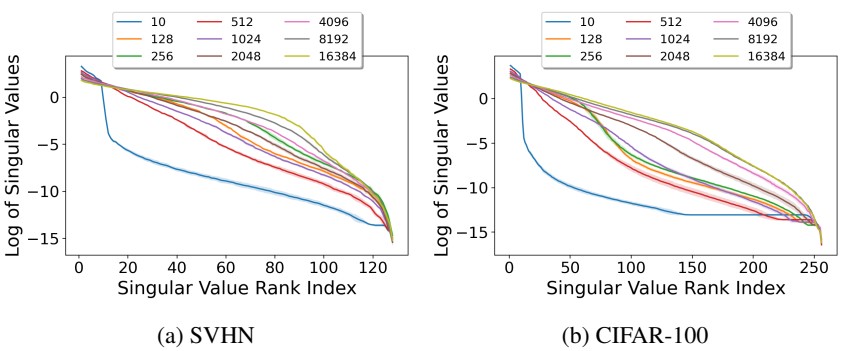

Figure 13: Dimensional collapse analysis on test data for different size of dictionary. Results are averaged over 5 training runs obtained from random initialization seeds.

from a smaller subset of values $\{0.1, 0.25, 0.5\}$ (given the more expensive nature of the experiments) to ensure both losses are minimized and chosen being equal to $0.25$.

**Evaluation.** For linear probe evaluation, we use the DINO codebase and train the classifier with Adam optimizer (Caron et al., 2021).

## L PRACTICAL IMPLEMENTATION OF THE LOSS

We observed training instability when using the larger backbone on ImageNet-100. The issue is due to some dictionary codes being unused during the initial training phase (cluster collapse), making the KL matching prior term infinity. Indeed, we have that

$$\mathcal{L}_{FALCON}(\mathcal{D}) = \beta CE(\boldsymbol{p}, \boldsymbol{p}') + CE(\boldsymbol{q}, \boldsymbol{p})$$
$$\propto \beta CE(\boldsymbol{p}, \boldsymbol{p}') + KL(\boldsymbol{q}, \boldsymbol{p})$$

In practice, the reverse KL term is sufficient to avoid the issue:

$$\mathcal{L}_{FALCON}(\mathcal{D}) = \beta CE(\boldsymbol{p}, \boldsymbol{p}') + KL(\boldsymbol{p}, \boldsymbol{q})$$

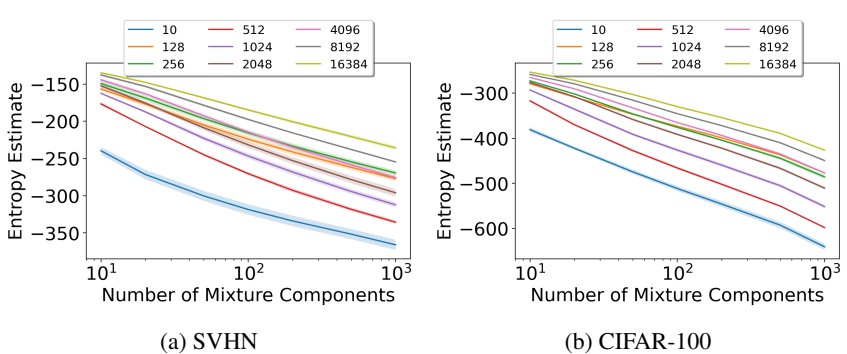

(a) SVHN                                         (b) CIFAR-100

Figure 14: Intracluster collapse analysis on test data for different size of dictionary. Results are averaged over 5 training runs obtained from random initialization seeds.