# OpenReview forum: "Failure-Proof Non-Contrastive Self-Supervised Learning"
_ICLR.cc/2025/Conference — Submitted to ICLR 2025_

### Official Review · Reviewer_tdwp · 2024-10-28

**Soundness:** 2
**Presentation:** 2
**Contribution:** 2
**Rating:** 5
**Confidence:** 3

**Summary:**

This paper proposed a method named FALCON to resolve known failure mode in non contrastive learning. Theoretically, the obtained representation are both decorrelated and clustered. They demonstrate the effectiveness of the method in several datasets.

**Strengths:**

1. This work introduces a novel approach to address known failure modes in non-contrastive learning. By identifying and tackling these specific limitations, the proposed method offers a valuable contribution to advancing the stability and accuracy of non-contrastive learning models.
2. The decision to draw W from a Rademacher distribution in the proposed method is a key innovation that enhances the FALCON loss. This modification may contribute to the regularization effect, thereby improving model generalization.

**Weaknesses:**

1. It is unclear how to select the hyperparameter $\beta$ in the loss function, which controls the contribution of specific components within the loss. Does optimal $\beta$ vary significantly across different datasets?
2. According to Lemma 1, we cannot infer that FALCON objective guarantee to avoid representation and cluster collapses. Lemma 1 shows that FALCON loss has a lower bound. The existence of a lower bound means that the loss cannot decrease indefinitely; however, it does not ensure that the model's learned representations will avoid collapsing to trivial solutions. Can the authors derive an upper bound for FALCON loss?
3. In the experiments, the accuracy of FALCON can outperform several methods but contrastive learning method can get better performances in these dataset. It is hard to convince readers to use FALCON loss. For example, a work: Revisiting a kNN-based Image Classification
System with High-capacity Storage (Kengo Nakata, et al.)

**Questions:**

1. One know failure mode is dimensional. How can FALCON scale up with high dimensional data? Especially how it can outperforms other methods.

---

> ### Author Response · Authors · 2024-11-28
> **Answers**
>
> Thank you for the valuable review. We have included detailed responses following each of your comments.
>
> **Choosing $\beta$.** $\beta$ is chosen to ensure that both the invariance and matching losses decrease throughout training. Currently, this choice is made manually, but a heuristic could be devised to adapt the hyperparameter and enforce this condition. However, we have opted to keep the approach as simple as possible. Regarding the variability of $\beta$, we did not observe significant differences in its choice across datasets. For instance, for large dictionary sizes, we use $\beta = 0.1$ on SVHN, CIFAR-10, and CIFAR-100, and $\beta = 0.25$ in all experiments on ImageNet100, with selections drawn from the grid $\\{0.01, 0.05, 0.1, 0.25, 0.5, 1, 2.5, 5, 10\\}$.
>
> **Lemma 1** Thank you for your question. Some clarifications are necessary. Having a lower bound that is computable in closed form is a desirable property of the objective function. This property ensures that attaining the optimal value provides a certificate for avoiding representation and cluster collapse. In practice, when using a finite-capacity or small-capacity backbone network, achieving the global optimum might not be feasible but avoidance can be still ensured. We refer you to Appendix B of the paper, where we have discussed this in more detail. Also, experimental results consistently confirm the avoidance of both representation and cluster collapse. For instance, see Fig. 6 for a conceptual explanation and Figs. 4d–f for further empirical evidence.
>
> **Weakness 3**. We would like to emphasize that the suggested paper considers a supervised setting, whereas our approach does not rely on labels and is therefore unsupervised. Additionally, the proposed solution is applied in a continual learning context, which is currently outside the scope of our work. Moreover, comparing with contrastive learning is not our primary goal; our aim is to bring together feature decorrelation and cluster-based approaches within the non-contrastive family (as also mentioned by reviewer an4Z).
>
> **Question** FALCON can outperform other methods thanks to the guarantees on generalization (see Appendix F). Intuitively (and less formally), this happens because the design of the projector and the loss function enforce feature decorrelation, thereby utilizing all neurons in the latent representation and preventing dimensional collapse. Having access to all neurons is crucial for generalization in downstream tasks, as it allows for the selection of relevant information needed to solve the task. The same argument does not apply when the representation is constrained to a smaller subspace, as this leads to a loss of information.
>
> We hope we've effectively addressed your concerns and encouraged a reconsideration of your rating. If you have any additional questions or feedback, please don't hesitate to reach out, we're happy to engage in further discussion.

---

### Official Review · Reviewer_KtxY · 2024-10-29

**Soundness:** 3
**Presentation:** 4
**Contribution:** 3
**Rating:** 6
**Confidence:** 4

**Summary:**

This paper pinpoints conditions to prevent common failures in non-contrastive self-supervised learning, such as representation and dimensional collapses. Using these insights, this paper develops a structured approach for the projector and loss function. The design inherently encourages learning representations that are both decorrelated and clustered, enhancing generalization without explicitly enforcing these traits. This appears to be the first method to ensure robust training against these failures while also improving generalization in subsequent tasks.

**Strengths:**

1. The writing logic of this paper is very clear, and the introduction part concisely explains the existing problems and the contributions of this paper.
2. This paper is the first to propose sufficient conditions to address the dimension and cluster collapse issues in non-contrastive self-supervised learning, making certain theoretical contributions to this field.
3. The figures are simple and clear, without unnecessary flourishes, which aid in understanding the FALCON method proposed in this paper.
4. The design of the loss function in Eq.2 is rational, and the role of the two terms in Eq.2 is clearly described. Moreover, based on the properties of Rademacher distribution, utilizing Rademacher distribution to avoid collapse is consistent with general intuition. Lemma 1, which concludes the existence and boundedness of global minima, also demonstrates the theoretical effectiveness of FALCON in avoiding collapse.
5. This paper presents a simple method that improves upon the state-of-the-art and is supported by solid theoretical foundations. It is a substantial and valuable contribution. The reviewer look forward to seeing the follow-up research mentioned in the conclusion by the authors in the future.

**Weaknesses:**

1. The author didn't provide source code in the supplement material.
Please refer to the question part.

**Questions:**

1. As the author mentioned, this paper proposes sufficient conditions to prevent potential mode collapses. The reviewer is concerned about one issue: if further considering the sufficient and necessary conditions to solve the collapse problem, would it be more helpful for future model design in this field? For example, it might be useful to consider narrowing down the function class where the projector resides and exploring what properties of the function class can prevent mode collapse.

---

> ### Author Response · Authors · 2024-11-28
> **Answers**
>
> Thank you for the thoughtful review and the positive feedback. Please find below some more detailed comments.
>
> **Code** The code will be publicly released upon acceptance.
>
> **Question** Thank you for the question and insightful comment ! We haven’t thought about this before but we can say that being able to constrain the function class will certainly facilitate and speed up training. This would be therefore a nice future avenue both from a theoretical as well as practical perspective. If you have any additional insights regarding this, we would be happy to discuss it further.

---

### Official Review · Reviewer_R44G · 2024-11-03

**Soundness:** 2
**Presentation:** 3
**Contribution:** 2
**Rating:** 5
**Confidence:** 4

**Summary:**

This paper proposed a non-contrastive self-supervised learning method that avoids critical failure modes.
1. It introduces a specially designed projector and loss function that decorrelates and clusters embeddings without ssl heuristics.
2. Provides theoretical guarantees for failure-proof training.
3. The method outperforms existing SSL methods on SVHN, CIFAR-10, CIFAR-100 and ImageNet-100 for discriminative tasks.

**Strengths:**

FALCON shows a theoretically grounded approach to non-contrastive SSL that avoids common collapse issues without standard heuristics. The paper is clear and well-organized, with intuitive figures and thorough validation across multiple datasets. It is the first paper that systematically deals with various failure modes and strengthens generalization at the same time.

**Weaknesses:**

Scalability Concerns: FALCON is not evaluated on the full ImageNet dataset, which limits comparison with standard ssl benchmarks. Running experiments on ImageNet would provide a stronger validation of FALCON’s scalability and practical performance.

Limited Training Duration: Most experiments are conducted over 100 epochs, which may not be sufficient for ssl methods to fully converge. Extending the training period could offer a more robust assessment of FALCON’s potential.

Similarity to Prior Work: The method shows resemblance to prior approaches like [1], particularly in using a uniform prior to prevent collapse (see Section 3.1 of the paper, “Why Degenerate Solutions are Avoided”). A clearer explanation of how FALCON differs, alongside comparisons with results from these prior methods, would strengthen the novelty claim.

[1] Ji, Xu, Joao F. Henriques, and Andrea Vedaldi. "Invariant information clustering for unsupervised image classification and segmentation." Proceedings of the IEEE/CVF international conference on computer vision. 2019.

**Questions:**

One common failure mode in discriminative self-supervised learning is the tendency for models to collapse to the color histogram of the input image, as addressed in SimCLR (see fig.6) and many subsequent works through the use of heavy data augmentation [1,2,3]. Have the authors considered such failure modes in their approach, and does FALCON have the potential to reduce reliance on data augmentations to avoid these issues?

[1] Chen, Ting, et al. "A simple framework for contrastive learning of visual representations." International conference on machine learning. PMLR, 2020.
[2] Caron, Mathilde, et al. "Unsupervised learning of visual features by contrasting cluster assignments." Advances in neural information processing systems 33 (2020): 9912-9924.
[3] He, Kaiming, et al. "Momentum contrast for unsupervised visual representation learning." Proceedings of the IEEE/CVF conference on computer vision and pattern recognition. 2020.

---

> ### Author Response · Authors · 2024-11-28
> **Answers**
>
> Thank you for the insightful review. Please, find below the answers to the different points.
>
> **Scalability** Thank you for this comment. We are aware of this, but we don’t currently have the resources to do the analysis on ImageNet1k. We have to rely on external resources to perform such analysis which cannot be achieved in a short timeframe. However, we believe that the theoretical contribution and experimental support of the paper brings new solid insights to the non-contrastive literature.
>
> **Limited Training** Thank you for the suggestion. Please refer to Table 2 in the revised paper where we have trained DINO and FALCON for 300 epochs. We report the table also here for reference
>
> | **Method**        | **128** | **1024** | **2048** | **4096** | **8192** | **16384** | **32768** | **65536** | **131072** |
> |-------------------|---------|----------|----------|----------|----------|-----------|-----------|-----------|------------|
> | **DINO (Top 1)**  | 71.8%   | 73.6%    | 73.9%    | 73.6%    | 74.3%    | 75.0%     | 75.1%     | 76.2%     | 75.8%      |
> | **FALCON (Top 1)**| **73.2%**| **74.2%**| **75.0%**| **76.3%**| **76.5%**| **76.9%**  | **77.5%**  | **78.1%**  | **77.1%**   |
> | **DINO (Top 5)**  | 92.1%   | **92.8%**| 92.9%    | 92.8%    | 93.0%    | 93.2%     | 93.0%     | 94.0%     | 94.0%      |
> | **FALCON (Top 5)**| **92.2%**| 92.4%    | **93.0%**| **93.2%**| **93.7%**| **94.0%**  | **94.3%**  | **94.2%**  | **94.5%**   |
>
>
> **Novelty** Thank you for pointing out the paper on Invariant Information Clustering (IIC). It is important to note that IIC and FALCON are conceptually different. IIC involves using a classification function on the backbone, and its objective is to maximize the mutual information between the class predictions of the two views. Consequently, there is a direct correspondence between prediction outputs and classes. In contrast, FALCON does not establish a one-to-one correspondence between codes and classes, as the dictionary can grow far beyond the number of classes and this is a key contribution of our work, establishing a new connection with hyperdimensional computing. Moreover, while the objective of IIC ensures the avoidance of degenerate solutions like full cluster collapse (i.e., collapsing to a single cluster), it does not prevent other types of failure. For example, as shown in Figure 3 (right) of the IIC paper, the learned solutions suffer from intra-cluster collapse. In FALCON, we avoid all failure modes, including the one mentioned above, due to the conceptual distinction between dictionary codes and clusters. We have cited and discussed the approach at L100 of the revised paper.
>
> **Question** Thank you for highlighting this important issue. Indeed, 'collapsing' to a set of low-level features such as color is another possible failure mode of non-contrastive learning methods. Since it can typically be avoided effectively using data augmentation, we did not pay much attention to this failure mode. We have added a note at L527 in the revised paper.
>
> We hope we've adequately addressed your concerns and encouraged you to reconsider your rating. If you have any further questions or feedback, please reach out, we’re open for further discussion.

---

> > ### Comment · Reviewer_R44G · 2024-12-01
> >
> > I thank the authors for their response. My concern regarding the scalability of the method has not yet been fully addressed. My current score has already factored in the theoretical contribution, so I am keeping the current score.

---

### Official Review · Reviewer_an4Z · 2024-11-03

**Soundness:** 4
**Presentation:** 3
**Contribution:** 3
**Rating:** 6
**Confidence:** 3

**Summary:**

The paper analyzes the failure modes of self-supervised learning—representation, dimensional, cluster, and intracluster collapses—and proposes a principled approach that is robust to these failure modes. The proposed approach, FALCON, maps feature representations to a frozen dictionary that follows a uniform distribution. The self-superivised learning loss is comprised of a invariance loss and a prior matching loss against the uniform distribution. Empirical results demonstrate competitive down-stream performance on SVHN, CIFAR-10, and CIFAR-100 datasets. Empirical results also demonstrate the benefits of increasing the size of the dictionary and training is more stable.

**Strengths:**

The proposed approach for self-supervised learning is simple and backed by a principled analysis that guarantees the proposed approach is robust to representation, dimensional, cluster, and intracluster collapses. I liked the idea of imposing a large uniform dictionary codes to facilitate representation learning, which brings together cluster-based SSL and feature decorrelation.

The empirical results show that the proposed FALCON with overcomplete dictionaries outperform all the baseline approaches and is more robust to collapses. The paper also carries out ablation to demonstrate the positive impact of increasing dictionary sizes.

**Weaknesses:**

Overall, I liked this paper from the motivation to the conclusions. It is unclear how to determine the hyperparameter \beta that balances the consistency and prior-matching loss and if the learning is robust to this hyperparameter. Also unclear if this approach would scale well with larger models.

**Questions:**

1. How to determine the hyperparameter \beta? Is learning robust to different values of \beta?
2. How would this approach scale with larger models? Would the size of dictionary create computational burden if c >> f?

--
Post-rebuttal:
I want to thank the authors for their response; I maintain my original rating.

---

> ### Author Response · Authors · 2024-11-28
> **Answers**
>
> Thank you for appreciating the soundness of our work ! Please find below the answers to your questions:
>
> **Setting $\beta$.** $\beta$ is chosen to ensure that both the invariance and matching losses decrease throughout training. Currently, the choice is made manually, but one could devise a heuristic to adapt the hyperparameter in order to enforce this condition. However, we have preferred to keep the approach as simple as possible. Regarding the sensitivity of the hyperparameter, we didn’t observe significant variations in performance for small (but non-zero) values of $\beta$. However, it is important to mention that when $\beta$ increases above approximately 0.5, performance decreases significantly. This is reasonable, as the uniformity loss starts to lose its importance, and codes are therefore not used uniformly.
>
> **Computation with larger models** As you correctly pointed out, the main bottleneck to scalability is the size of the dictionary. There are a few considerations in this regard that could potentially lead to follow-up work. First, while we have a guarantee that performance improves with larger dictionaries, it is important to note that the improvement is not strictly monotonic, and performance plateaus after a certain point (see, for instance, Table 2). Consequently, we don’t need to increase the dictionary size indefinitely. Second, we could better exploit the structure of the dictionary. Currently, we are storing $f\times c$ floats, but one could exploit the bipolar nature of the tensor and develop a more efficient way to store and use it during matrix multiplication. We have not explored this direction yet. Thanks for raising the question ! We have added a paragraph at L530 in the revised paper.
>
> Please let us know if you have any additional questions. We would be happy to answer them !

---

### Official Review · Reviewer_zgjG · 2024-11-04

**Soundness:** 3
**Presentation:** 2
**Contribution:** 3
**Rating:** 6
**Confidence:** 2

**Summary:**

This paper introduces a new non-contrastive loss function as well as projector layer design, motivated by theory to avoid the known problems of representation collapse, dimensional collapse and intra-cluster collapse for these methods. Previous methods have provided more complicated losses (e.g. generative loss) or heuristics (e.g. stop gradient, momentum) to achieve this, but this paper proposes an alternative by modifying the projector layer and the loss function. The theoretical contributions are backed with experiments to show that the proposed method FALCON can outperform prior methods in generalization while avoiding the known problems in a principled manner.

**Strengths:**

- The problem is very well motivated. It is true that non-contrastive methods have been relying on heuristic solutions to deal with the different types of collapse problems
- The paper provided theoretical justification for the FALCON loss and projector design
- This is also supported by experiments showing not just the performance, but the distribution of representations to show how the method avoids collapse.

**Weaknesses:**

Personally, I found the theory section extremely difficult to parse carefully. There is far too many technical expressions before the problem is clearly formulated and insufficient explanation to understand the key ideas of the proof.

As an example, the presentation of Lemma 1 is inscrutable. It is hard to know 1) what does the extrema condition intuitively mean? 2) On line 227, the authors ask the reader to note that the invariance loss can be decomposed into a "sum of entropy" term and a "KL-divergence" term -> this is not easy to do without the authors showing this decomposition.

Overall, I feel this paper can benefit greatly from rewriting to present the technical content in a more digestible manner.

**Questions:**

Covered in weaknesses.

---

> ### Author Response · Authors · 2024-11-28
> **Answers**
>
> Thank you for the time you dedicated to understanding and reviewing our paper. We are happy to make the paper more accessible. Please, find below some clarifications to your questions.
>
> **Extrema condition** To gain some intuition, consider the case with $\epsilon=0$. That condition translates to have one of the p values equal to 1 and the others equal to 0. This can be thought of reaching the extrema of a simplex. We can add a sentence after Lemma 1 to reflect that the notion of extremum is referred to the extremum of a probability simplex. We have added a sentence at L213 in the revised paper.
>
> **Decomposition of invariance loss** The invariance loss in Eq. 2 can be expressed as a cross-entropy loss. The mentioned decomposition follows directly from the property of the cross-entropy loss, that is $CE(p,p’)=H(p)+KL(p\|p’)$. We have added a clarifying sentence at L228 in the revised paper.
>
> We hope the above explanations, along with the new clarifications in the paper, enhance its understanding. If you have any further questions or feedback, we are happy to engage in discussion.

---

### Official Review · Reviewer_Za7r · 2024-11-04

**Soundness:** 3
**Presentation:** 3
**Contribution:** 3
**Rating:** 6
**Confidence:** 4

**Summary:**

The paper introduces FALCON, a clustering based self-supervised approach that tackles various kinds of collapses through architectural or loss related changes. This allows it to avoid classical collapsing behaviour (total/dimensional) but also collapses related to clustering.
The author demonstrate that this non-collapsing behaviour is avoided both theoretically and empirically, proving that these problems are indeed alleviated.
This work thus extends dimensional collapse analysis to clustering based methods while proposing a method avoiding all of these collapsing behaviours, leading to increased performance.

**Strengths:**

- The in depth analysis of clustering related collapse is very interesting as cluster based methods have gained in popularity recently (e.g. DINOv2). This extends previous work on dimensional collapse focusing primarily on dimensional collapse.
- Having both the theoretical and empirical evidence that FALCON effectively avoids diverse kinds of collapse leads to a convincing approach.
- FALCON demonstrates improved performance over the baseline methods considered (with some caveats, see weaknesses)

**Weaknesses:**

The experiments appear a bit weak to support the claims of improved performance.

For the experiments, the use of ResNet-8 leads to extremely low performance for all methods considered (At least for CIFAR-10/100). Comparing to a more standard ResNet-18 (see https://github.com/vturrisi/solo-learn?tab=readme-ov-file#cifar-10 for performance for various methods) we see gaps of 20-30 points. It’s thus unclear how these results may transfer to more widely used architectures.The same analysis on ResNet-18 backbones would be more convincing.
For ImageNet-100, it would also be beneficial to add similar baselines as Table 1 (e.g. Barlow Twins, Swav) to understand how FALCON relates to other methods in this larger scale setting.

**Questions:**

- Lines 169-170, key components are described as a BN then L2-norm. This seems very similar to doing one step of the Sinkhorn-Knopp algorithm. Was this the motivation for the predictor design ? It may be worth discussing this link more to give an intuition as to how this design helps avoid cluster collapse (due to the sinkhorn knopp algortihm helping to spread cluster assignments)

- Regarding the description of collapse types, a type that exists in methods such as DINO is that some prototypes become identical, and thus we get a kind of “prototype collapse” (see appendix C in [1]), where even if we define $n$ prototypes, there are only $n<m$ in practice. This is directly visible as it is identical to dimensional collapse for clustering methods (since each dimension after clustering corresponds to a dimension). This can for example be avoided by decorrelating features (and thus clustering prototypes). It may be good to discuss the link between this type of collapse and dimensional collapse when discussing all kinds of collapse.

- For the comparison to DINO on dictionary size in table 2, what are the “actual” number of prototypes (and thus dimensions) as the dictionary size increases ? Do we see a stronger collapse for DINO which FALCON avoids ? an analysis such as done in appendix C of [1] may help demonstrate clearly that FALCON does a better job at using the clustering than DINO.

Notation:

- In section 3, $f$ is used as the dimension of the representations and $g$ to denote the encoder. As $f$ is usually used for the encoder, another choice for the embedding dimension (e.g. $d$ and using something like $d’$ for the input dimension) may clarify the notation.

[1]Garrido, Q., Balestriero, R., Najman, L., & Lecun, Y. . Rankme: Assessing the downstream performance of pretrained self-supervised representations by their rank. ICML 2023.

---

> ### Author Response · Authors · 2024-11-28
> **Answers**
>
> Thank you for the appreciation of our work and for the suggestions to improve the paper. Please find below more details about the points you mentioned.
>
> **Additional Experiments** Due to constraints in time and computational resources, we were unable to address all reviewers' requests. We conducted additional experiments with extended training epochs specifically for reviewer R44G.
>
> **Batch norm and L2 normalization** Thank you for pointing out this intuition. While it is true that both batch normalization (BN) and the L2 norm aid in optimization, they serve different purposes. BN helps condition the optimization to ensure good statistics (zero mean and avoiding variance explosion, as shown in Appendix A). Zero mean is necessary to ensure that the data is spread out on the hypersphere, as you correctly pointed out. However, the L2 norm is crucial for constraining the admissible optimal values of $\alpha$ (as shown in the proof of Theorem 1, L1009–L1024). In other words, the L2 norm enforces a sparse assignment of data to codes. We have added a sentence at L181 in the revised version of the paper.
>
> **Prototype collapse** Thank you for pointing out the weakness of DINO and referring us to Appendix C of that paper. While this phenomenon is present in DINO, it does not occur in our settings, as the codes are frozen throughout the learning process. Analyzing the singular value distribution directly on the codes of FALCON would be trivial due to the frozen dictionary. We have modified L513 in the revised paper to mention this. This is a nice touch !
>
> **Notation** Thanks for the suggestion. In our opinion, the apex in $d’$ could overload the notation as it is used for augmentations. $f$ stands for features. While this is an arguable convention, we would prefer to maintain the current notation.
>
> We hope we have effectively addressed your concerns, enhancing the paper's value from your perspective. Should you have any additional feedback, we would be glad to discuss it further.

---

> > ### Comment · Reviewer_Za7r · 2024-12-02
> >
> > I thank the authors for their answers to my questions. While I have an overall positive opinion about the paper and the contributions, I will keep my score as I believe that the experimental evaluation remains on the weaker side.

---

### Author Response · Authors · 2024-11-28
**General Comment**

We would like to thank all the reviewers for their valuable suggestions to improve our paper and for their overall appreciation of our work. We have incorporated these suggestions into the revised version of the paper. We apologize for the delayed response, as we encountered some issues with the computational resources needed to run additional experiments. We have provided detailed answers to each review individually and remain available for further discussion.

---

### Author Response · Authors · 2024-12-03
**Conclusion of discussion phase**

Dear reviewers,

Thank you once again for your suggestions aimed at valorising the theory and improving our paper.
Please let us know if there is anything else you would like to discuss during the remaining phase of the rebuttal.

The Authors

---

### Meta-Review · Area_Chair_JTrw · 2024-12-19

**Metareview:**

The paper studies known failure modes of self-supervised learning approaches and proposes a method that has an inductive bias that avoids representation collapse.
The reviewers argued that the problem is interesting and a theoretically sound solution would be very useful to the community.
- Unfortunately, the presentation of the paper was found to be insufficient and the reviewers recommend a rewrite before the paper is accepted.
- I also agree that the experiments with ResNet8 are too low performance and it is unclear how the method will fare in larger scale settings.

**Additional Comments On Reviewer Discussion:**

The discussion was active, and the reviewers were not satisfied with the author's reply, so they decided to keep their scores and ultimately supported rejection.

---

### Decision · Program_Chairs · 2025-01-22

Reject